# Optical control of resonances in temporally symmetry-broken metasurfaces

Andreas Aigner[1,5], Thomas Possmayer[1,5], Thomas Weber[1], Alexander A. Antonov[1], Leonardo de S. Menezes[1,2], Stefan A. Maier[3,4✉] & Andreas Tittl[1✉]

Tunability in active metasurfaces has mainly relied on shifting the resonance wavelength[1,2] or increasing material losses[3,4] to spectrally detune or quench resonant modes, respectively. However, both methods face fundamental limitations, such as a limited $Q$ factor and near-field enhancement control and the inability to achieve resonance on–off switching by completely coupling and decoupling the mode from the far field. Here we demonstrate temporal symmetry breaking in metasurfaces through ultrafast optical pumping, providing an experimental realization of radiative-loss-driven resonance tuning, allowing resonance creation, annihilation, broadening and sharpening. To enable this temporal control, we introduce restored symmetry-protected bound states in the continuum. Even though their unit cells are geometrically asymmetric, coupling to the radiation continuum remains fully suppressed, which, in this work, is achieved by two equally strong antisymmetric dipoles. By using selective Mie-resonant pumping in parts of these unit cells, we can modify their dipole balance to create or annihilate resonances as well as tune the linewidth, amplitude and near-field enhancement, leading to potential applications in optical and quantum communications, time crystals and photonic circuits.

Active nanophotonics is a rapidly advancing field that offers promising solutions for many emerging technologies, like holography, quantum cryptography and optical computing[5,6]. In particular, active metasurfaces, two-dimensional arrays of subwavelength-spaced nanoresonators, have emerged as a powerful tool for manipulating and confining light[7,8]. They have been successfully applied in beam steering[9,10], optical switching[11,12], holography[13], adjustable lenses[13,14], tunable sensors[15,16], programmable surfaces[17,18], and active chiral[19] and polarization[20] filters. In general, the tunability of a resonant system, as shown in Fig. 1a, is achieved by altering one or more of the fundamental resonance parameters: resonance wavelength $\omega_0$, intrinsic loss $\gamma_{int}$ and radiative loss $\gamma_{rad}$, with the literature predominantly focusing on $\omega_0$ (refs. 1,2,21–23) and $\gamma_{int}$ (refs. 3,4,11,24,25). Tuning $\omega_0$ shifts the resonance spectrally, whereas tuning $\gamma_{int}$ dampens the resonant mode, both resulting in changes to the amplitude at specific wavelengths. The idealized cases of these two tuning methods are illustrated in Fig. 1b,c, respectively. In reality, however, these parameters are intertwined due to the Kramers–Kronig[26] relations linking the real part ($n$) and the imaginary ($k$) of the refractive index: $\omega_0$ is primarily influenced by $n$ and $\gamma_{int}$ by $k$. Although both tuning methods have achieved notable results, they face inherent limitations. These limitations become evident when examining the far-field response of a single photonic mode, here described by the Lorentzian transmission coefficient[27] (Fig. 1a):

$$t(\omega) = 1 - \frac{\gamma_{rad}}{i(\omega - \omega_0) + \gamma_{rad} + \gamma_{int}}. \qquad (1)$$

It is well known from the literature that both $\omega_0$ and $\gamma_{int}$ can influence the amplitude and linewidth of the modes and, thus, also the local field enhancement. However, neither can truly turn the mode on or off by changing the fraction term in equation (1) from zero to a finite value or vice versa. Such control requires coupling (or decoupling) of the mode to the radiation continuum, effectively toggling the mode between bright and dark states, which is mediated by the third resonance parameter, $\gamma_{rad}$. A $\gamma_{rad}$ of zero means that there is a mode fully decoupled from the far field, rendering it off, whereas $\gamma_{rad} > 0$ switches the mode on, both sketched in Fig. 1d. This capability is transformative for active photonics, particularly in applications like optical communications, signal processing and filtering, where it can minimize spectral crosstalk and avoids parasitic losses. Furthermore, in contrast to $\omega_0$ and $\gamma_{int}$, $\gamma_{rad}$ directly governs the resonance amplitude, $Q$ factor and local field enhancement, making it critical for precise control of metasurface functionalities. Despite its great potential, achieving active control of $\gamma_{rad}$ remains challenging because simple modifications of the refractive index are not sufficient to tune $\gamma_{rad}$. Even in passive metasurfaces, controlling $\gamma_{rad}$ has only recently become feasible with the introduction of symmetry-protected bound states in the continuum (SP-BICs)[28,29]. These states enable passive $\gamma_{rad}$ control by breaking the geometric symmetry in a metasurface unit cell[30]. However, for the first active SP-BICs, the focus has been on tuning $\omega_0$ (refs. 19,31–35) or $\gamma_{int}$ (refs. 15,25,31,36,37), without fully exploiting the unique ability of SP-BICs to adjust $\gamma_{rad}$. Initial experimental efforts to tune $\gamma_{rad}$ have been hindered by substantial intrinsic losses[15,38], such that the $Q$ factor

[1]Chair in Hybrid Nanosystems, Faculty of Physics, Ludwig-Maximilians-University Munich, Munich, Germany. [2]Departamento de Física, Universidade Federal de Pernambuco, Recife-PE, Brazil. [3]School of Physics and Astronomy, Monash University, Clayton, Victoria, Australia. [4]Department of Physics, Imperial College London, London, UK. [5]These authors contributed equally: Andreas Aigner, Thomas Possmayer. ✉e-mail: stefan.maier@monash.edu; andreas.tittl@physik.uni-muenchen.de

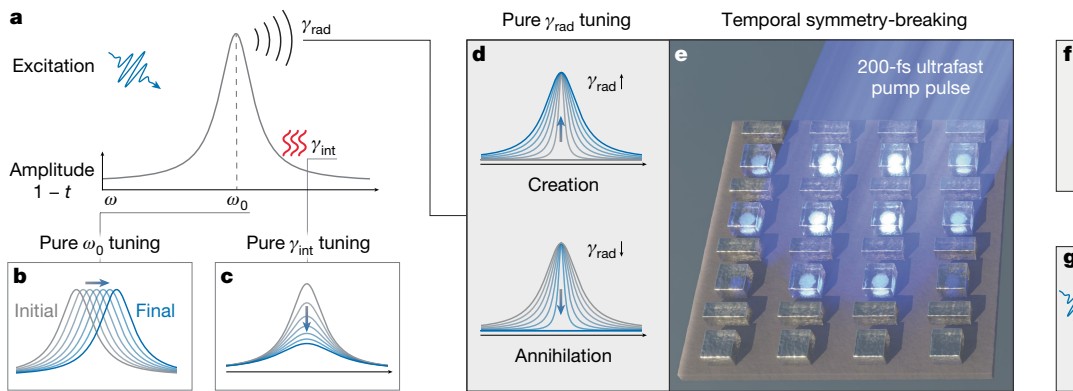

**Fig. 1 | Active tuning mechanisms and control of ultrafast radiative loss through selective pumping. a**, Illustration of an initial resonant mode before excitation using $1 - t(\omega)$. As indicated, the mode is defined by three parameters: $\omega_0$, $\gamma_{int}$ and $\gamma_{rad}$. **b–d**, Active tuning approaches for $\omega_0$ (**b**), $\gamma_{int}$ (**c**) and $\gamma_{rad}$ (**d**). The grey curves are the initial resonances, and the blue curves are the tuned resonances. **b**, Tuning $\omega_0$ results in a shifted resonance profile although the amplitude and spectral width remain unchanged. **c**, Tuning $\gamma_{int}$ quenches the amplitude of the resonance without fully eliminating it. **d**, Tuning $\gamma_{rad}$ allows the mode to be created as $\gamma_{rad}$ increases from zero and fully annihilated as $\gamma_{rad}$ decreases to zero. **e**, Illustration of the temporal symmetry-breaking metasurface, selectively pumped with a 200-fs pulse resonant with a Mie mode in only one of two rods per unit cell indicated by the glowing areas. **f**, The metasurface initially exhibits an RSP-BIC. The dipole moments of both rods, which have refractive indices $n_{rod1} = n_{rod2}$, are of equal strength, resulting in an almost perfect antisymmetric mode profile with $\gamma_{rad}$ approaching zero. **g**, After resonant absorption of the pump pulse, the refractive index in rod 1 decreases ($n_{rod1} < n_{rod2}$), resulting in asymmetric dipole moments and the emergence of the quasi-SP-BIC mode ($\gamma_{rad} \neq 0$).

was changed only to a limited degree. Although a recent numerical study investigated boosting the $Q$ factor through interference-based pumping[39], the ability to arbitrarily tune the radiative loss to increase or decrease the $Q$ factor and achieve on–off switching has so far not been realized.

To address these challenges, we present a new experimental approach that demonstrates temporal symmetry breaking and $\gamma_{rad}$ tuning in metasurfaces using collinear ultrafast optical pumping (Fig. 1e). This enables the creation and annihilation of high $Q$-factor resonances on a subpicosecond timescale. The resonator material is modulated through photo-excited charge carriers as a first proof of concept. The temporal tuning of $\gamma_{rad}$ is made possible by leveraging restored SP-BICs (RSP-BICs), which are experimentally introduced in this work. Here the structural symmetry within a unit cell is broken, but for light at a specific wavelength, the system behaves symmetrically with cancelling of antiparallel dipoles, and hence, $\gamma_{rad}$ approaches zero. In our work, the unit cells consist of two rods of different lengths and widths (Fig. 1f) that exhibit equal dipole moments at the RSP-BIC wavelength. However, for other wavelengths and polarizations, each has a set of unique Mie modes. This enables us to lower the refractive index $n$ in only one of the two rods, which we photo-excite selectively by resonantly pumping its Mie mode. The change in $n$ alters the ratio of the individual dipole moments (Fig. 1g), thus modifying their asymmetry and $\gamma_{rad}$ of the metasurfaces[40]. In transient absorption experiments, we achieve ultrafast control of $\gamma_{rad}$ near the RSP-BIC condition in four ways: we can sharpen, broaden, create or annihilate resonances depending on the geometry of the system and the pump fluence, whereas the effect on $\gamma_{int}$ is negligible.

## Restored symmetry-protected BICs

Conventional SP-BICs rely on in-plane geometric inversion symmetry within the unit cell to suppress coupling to radiative channels. Fundamentally, however, coupling is prevented if the effective dipole moment of the unit cell is zero. Notably, we demonstrate that it is possible to break the geometric symmetry of the resonators while maintaining a near-zero effective dipole moment: the RSP-BIC condition with $\gamma_{rad} \approx 0$.

We use crystalline silicon as resonator material due to its optical tunability[38,41,42] and fairly low losses at the target wavelength of 800 nm. The nanoresonators, which have a height of 115 nm, were fabricated on a sapphire substrate and encapsulated with silicon dioxide ($SiO_2$). Our chosen geometry comprises two aligned dipolar rods within a $420 \times 420$ nm$^2$ unit cell (Fig. 2a). When their respective lengths and widths match, the system exhibits symmetry protection: the opposing dipole moments cancel ($p_{tot} = 0$), and the mode is decoupled from the far field, resulting in a vanishing $\gamma_{rad}$ (Fig. 2a). Breaking the symmetry by increasing the width $w_1$ of the first resonator increases its dipole moment. This yields a net asymmetry of the combined structure, and a quasi-BIC forms with $p_{tot} > 0$ (Fig. 2b). Next, we increase the length $l_2$ of the second resonator, thus increasing its dipole moment. At a specific combination of $w_1$ and $l_2$, the dipole moments closely match again ($p_{tot} \approx 0$), which restores the photonic symmetry despite the broken geometric symmetry. This is the RSP-BIC condition, where $\gamma_{rad}$ returns to zero, and the radiative $Q$ factor diverges (Fig. 2c).

In the simulations (Methods), the initial resonator lengths and widths are set to 175 nm and 95 nm, respectively. Figure 2d shows the emergence of the SP-BIC mode around 770 nm when $w_1$ increases. The fitted $\gamma_{rad}$ in Fig. 2e (TCMT model in Supplementary Note 1) converges to zero around the symmetric case, which is typical for SP-BICs. Continuing with the asymmetric case ($w_1 = 185$ nm), increasing $l_2$ (right panel of Fig. 2d) causes the mode to sharpen and eventually disappear at $l_2 = 216$ nm, which corresponds to the RSP-BIC condition with $\gamma_{rad} \approx 0$ (right panel of Fig. 2e). For even larger $l_2$, the mode reappears. For all cases, Supplementary Note 2 shows good agreement with $Q_{rad} \propto 1/\alpha^2$ before and after the RSP-BIC condition and reveals an identical mode profile for all $l_2$, confirming the SP-BIC nature of both branches.

Based on these numerical results, we fabricated nanoresonators that match the simulated designs (see Methods and the workflow in Extended Data Fig. 1). Scanning electron microscopy (SEM) images of the unit cells before $SiO_2$ encapsulation are shown in Fig. 2f for the SP-BIC, quasi-SP-BIC and RSP-BIC conditions. To ensure a continuous transition between these states, we design two gradient metasurfaces[43] mimicking the numerical results shown in Fig. 1d, each gradient with a size of $100 \times 50$ μm$^2$ (Methods). In the first, $w_1$ continuously increases from 95 nm to 185 nm for constant $l_2 = 175$ nm, and the second has $w_1 = 185$ nm and $l_2$ increases from 175 nm to 275 nm. A true colour optical image is shown in Fig. 2g. The gradual colour change indicates the smooth variation of $w_1$ and $l_2$ (SEM images and line spectra shown in Extended Data Figs. 2 and 3, respectively). The experimental transmittance spectra in Fig. 2h, extracted along the $x$ axis (Methods), confirm

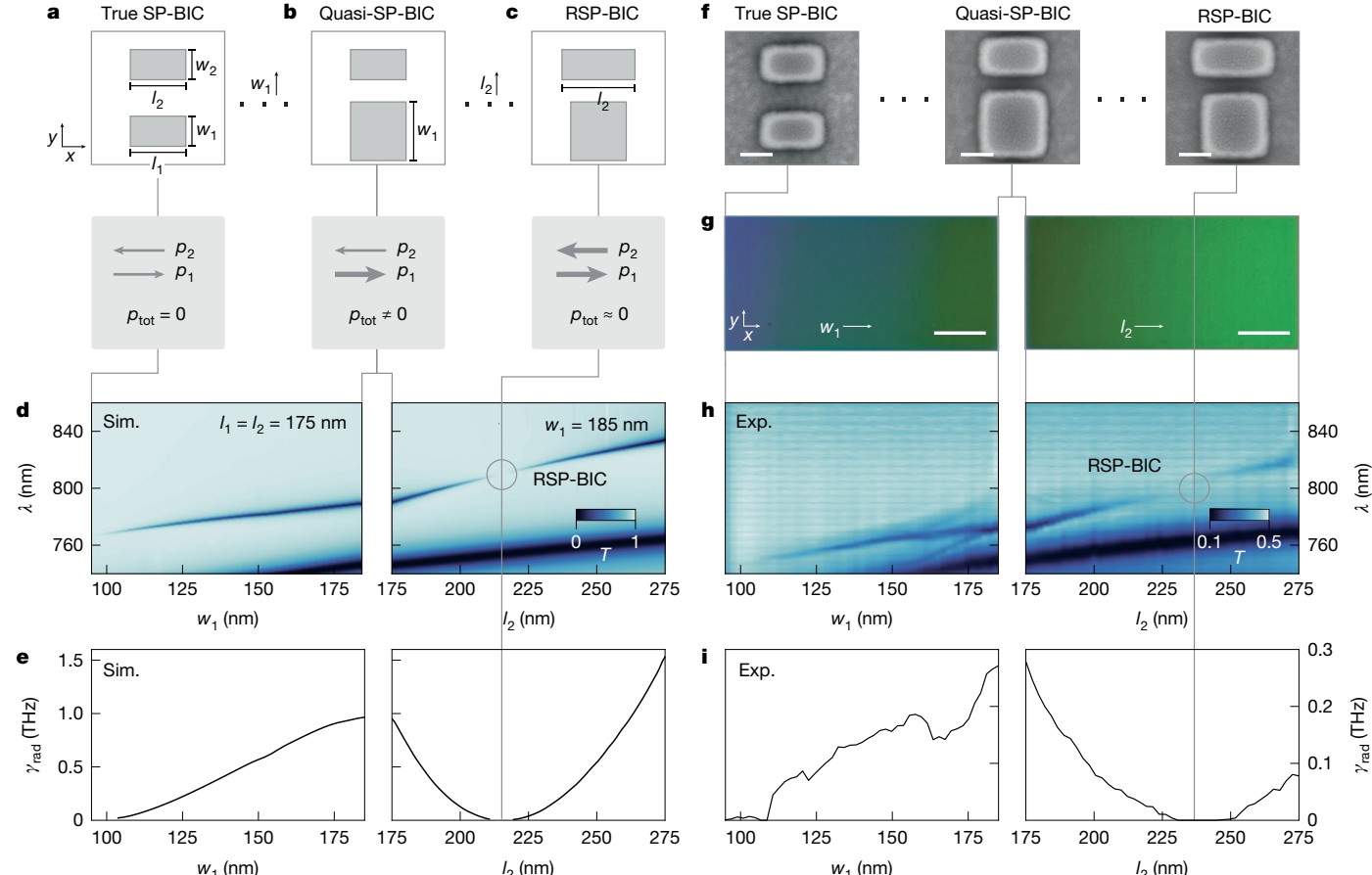

**Fig. 2 | RSP-BIC principle and experimental verification. a**, Sketch of the unit cell geometry, which consists of two crystalline silicon rods with lengths $l_1$ and $l_2$ and widths $w_1$ and $w_2$, respectively. The dipole moments $p_1$ and $p_2$ along the $x$ axis are equal, resulting in a total dipole moment $p_{tot} = 0$ for an out-of-phase mode, indicating a SP-BIC condition. **b**, When $w_1$ is increased, the symmetry is broken (quasi-BIC), and $p_{tot} \neq 0$, allowing the mode to couple to the far field. **c**, Increasing $l_2$ restores the symmetry, returning the system to $p_{tot} \approx 0$, the RSP-BIC condition. **d**, Numerical transmittance spectra of the SP-BIC mode as $w_1$ is varied from 95 nm to 185 nm (left). Tuning $l_2$ from 175 nm to 275 nm for fixed

$w_1 = 185$ nm sharpens the mode until it disappears at the RSP-BIC (marked by the grey circle) (right). **e**, $\gamma_{rad}$, obtained from TCMT fitting, converges to zero at the SP-BIC and RSP-BIC conditions. **f**, SEM images of the crystalline silicon metasurface corresponding to the cases shown in **a**–**c**. **g**, Optical images of the two gradient metasurfaces. Left, a $w_1$ gradient. Right, an $l_2$ gradient. **h**, Experimental spectra matching the numerical results in **d**. **i**, Fitted experimental data for $\gamma_{rad}$, corresponding to the results in **e**. Exp., experimental; Sim., simulated. Scale bars, 50 nm (**f**), 20 μm (**g**).

the presence of the same SP-BIC mode as seen in Fig. 2d. The small spectral shift between the simulated and experimental results of around 10 nm can be attributed to slight geometrical offsets between the simulated and measured geometries. For the gradient on the right, the RSP-BIC condition is evident in Fig. 2i as $\gamma_{rad}$ converges to zero before reappearing. The seamless tracing of this mode back to the conventional SP-BIC, in both simulations and experiments, strongly supports the symmetry-protected nature of the RSP-BIC condition. It is important to note that even though we see vanishing radiative damping and transmittance signal for the RSP-BIC, the $Q$ factor does not go to infinity but takes on a finite (but very large) value above $10^7$ in the simulations (Supplementary Note 3).

## Selective optical pumping

After experimentally verifying the RSP-BIC condition with $\gamma_{rad} \approx 0$ in a highly asymmetric unit cell, we now investigate broadband transmittance spectra (Fig. 3a) to find other optical modes of the system that could be used for selective optical pumping. Although the RSP-BIC mode is not visible here due to $\gamma_{rad} \approx 0$, three distinct dips can be seen, which we attribute to Mie modes: two for $y$-polarized light at 720 nm and 745 nm, labelled Mie 1 and Mie 2, respectively, and one for $x$-polarized

light at 764 nm labelled Mie 3. Based on multipole decompositions (Supplementary Note 4), we can assign an electric dipole-like behaviour to Mie 1, whereas Mie 2 and Mie 3 are magnetic dipole-like modes.

As the active tunability is based on lowering $n$ through photo-excitation, we compare the distributions of the power loss density (energy loss per unit volume in W m$^{-3}$) for both rods. Figure 3b shows the simulated power loss density profile for all three Mie modes cut at a height of 30 nm. Strikingly, we observe higher power loss densities in rod 1 for all three modes (Fig. 3c). Mie 1 exhibits a particularly strong imbalance with a 3.5-fold higher absorption per volume in rod 1 than in rod 2. Since a high absorption imbalance yields a high $n$ imbalance, we select Mie 1 as the ideal mode for efficient $\gamma_{rad}$ tuning.

Our tuning approach is based on the above-bandgap pumping of silicon, sketched in Fig. 3d. This modifies the polarizability of the material, lowering the refractive index[44] from $n_{Si,0}$ to an absorption-dependent $n_{Si,Pump}$. The decrease occurs on the timescale of the pump pulse (here 200 fs), and its recovery is mainly defined by carrier recombination through three mechanisms: surface, trap and Auger recombination. Owing to the high surface-to-volume ratio of our structure and the defect density caused by nanofabrication, the recombination is expected to happen significantly faster than in bulk silicon. We use a geometry with $l_2 \approx 236$ nm to ensure that the SP-BIC mode is

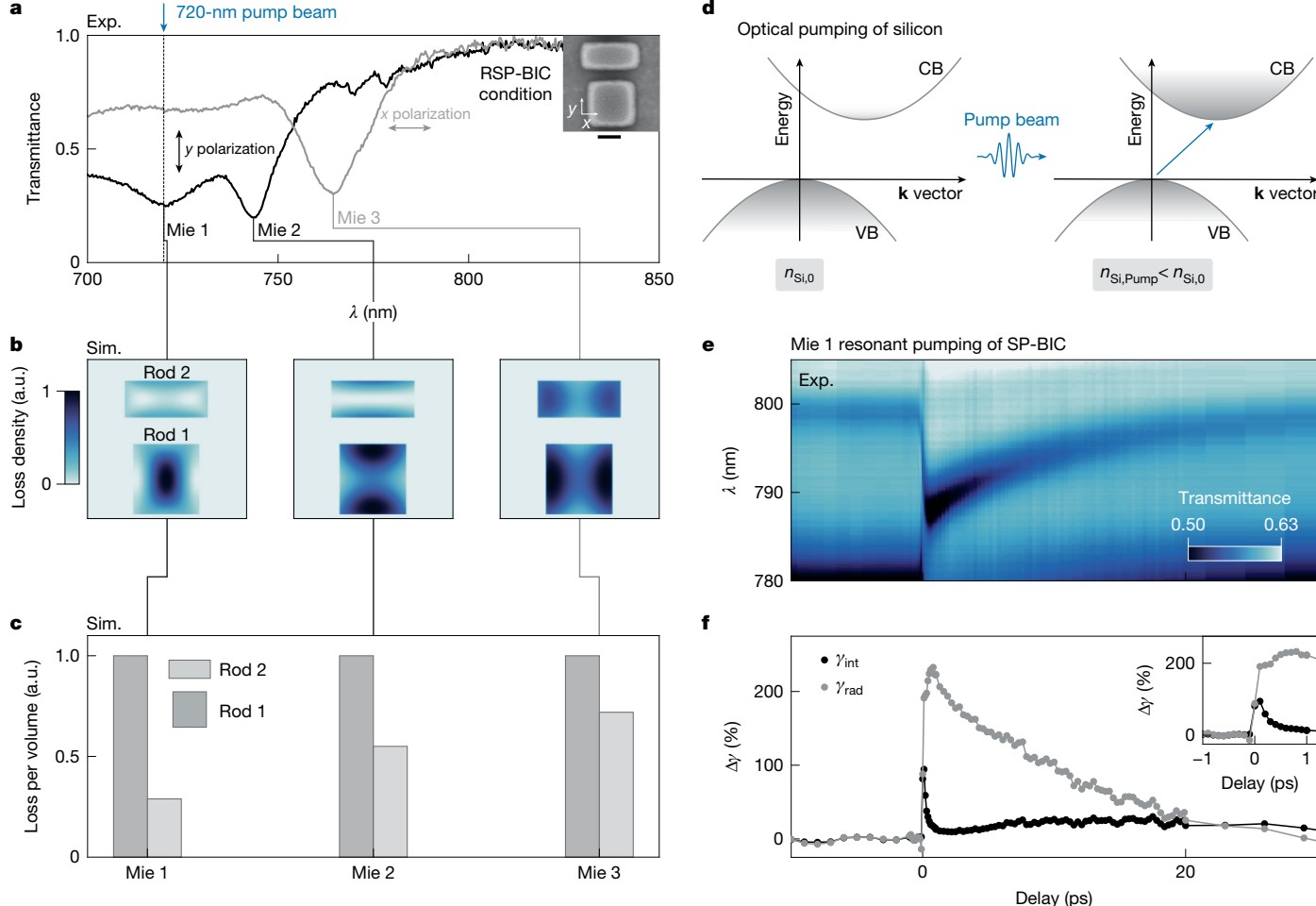

**Fig. 3 | Mie modes and selective resonant pumping. a**, Experimental transmittance spectrum at the RSP-BIC condition with $l_2 \approx 226$ nm for $x$- and $y$-polarized light, revealing two and one Mie modes, respectively (labelled Mie 1, Mie 2 and Mie 3). **b**, Normalized power loss density map for the unit cells (energy loss per time and unit volume) for the three Mie modes shows distinct mode profiles and dissimilar losses in both rods at a cutting plane of $z = 30$ nm. **c**, Comparison of average power loss density between the two rods for each Mie mode. Mie 1 exhibits the highest ratio between rods 1 and 2, indicating the strongest selective absorption. **d**, Sketch of the optical pumping principle. The above-bandgap excitation of carriers from the conduction band (CB) to the valence band (VB) generates free electrons and holes, which alters the polarizability and the refractive index. **e**, Pump–probe spectral time trace (720-nm pump with a fluence of 100 µJ cm⁻²) for a metasurface with $l_2 = 236$ nm. The pump pulse is $y$-polarized and the probe pulse is $x$-polarized, leading to a 9.2-nm spectral shift of the SP-BIC and an increase in the resonance amplitude. **f**, Corresponding $\gamma_{int}$ and $\gamma_{rad}$ obtained using TCMT fitting. A sharp increase of $\gamma_{int}$ by approximately 100% upon pump arrival, followed by a rapid drop to approximately 14% above pre-pump values within 1 ps is visible. $\gamma_{rad}$ increases by 250%, remaining at higher values before it returns back to pre-pump values within 20 ps. Inset, the quick decay of intrinsic losses within the first picoseconds after pump arrival. Scale bar, 100 nm (**a**).

visible, while the same Mie modes as in the RSP-BIC case are still present (Extended Data Fig. 4). To excite Mie 1, we pump the sample with $y$-polarized light at a wavelength of 720 nm and a fluence of 100 µJ cm⁻² (the pump–probe set-up is sketched in Extended Data Fig. 5). A broadband probe pulse monitors changes in the transmission at variable delay times. Non-resonant ($x$-polarized) and spectrally detuned pumping, shown in Extended Data Fig. 6 and Supplementary Note 5, respectively, both lead to only minor spectral shifts with an otherwise unchanged resonance profile. By contrast, Fig. 3e shows a time trace for a 720-nm $y$-polarized pump pulse on resonance with Mie 1. A spectral shift of 9.2 nm is observed, which decays to the original spectral position with a time constant of around 20 ps. Furthermore, after the pump, the resonance amplitude shows a clear increase, indicating a significant change in $\gamma_{rad}$ due to the experiments being performed in the undercoupled regime. Using the TCMT model (Supplementary Note 1), we fit the transient spectra (Supplementary Note 6) and extract the time-dependent loss rates shown in Fig. 3f. Note that the initial value of $\gamma_{int}$ has several contributions, including material, surface roughness and finite array size losses. Around 1 ps after the pump, $\gamma_{int}$ and $\gamma_{rad}$ increase by roughly 14% and 250%, respectively, and decay within 20 ps. In absolute terms, $\gamma_{int}$ increases from around 0.9 to 1.0 THz and $\gamma_{rad}$ increases from 0.04 to 0.14 THz. In addition to the carrier concentration, $\gamma_{int}$ is also susceptible to two-photon absorption, electron temperature and mode shifting, leading to deviating decay behaviour from $\gamma_{rad}$ (Supplementary Note 7). Because the sustained modulation is dominated by $\gamma_{rad}$, the following analysis focuses on modelling the radiative loss.

## Refractive-index perturbation and ultrafast tuning

To quantify the pump-induced change in the refractive index, we match the experimentally observed resonance shift of 9.2 nm to numerical simulations (Supplementary Note 8), yielding a refractive-index difference between the two rods of $\Delta n = n_{rod2} - n_{rod1} = 0.13$ (Fig. 4a). We employ resonant-state expansion (RSE) theory[45] to interpret this value analytically (Supplementary Note 3). For simplicity, we assume a constant crystalline-silicon index $n_0 = 3.7$ and place the metasurface in

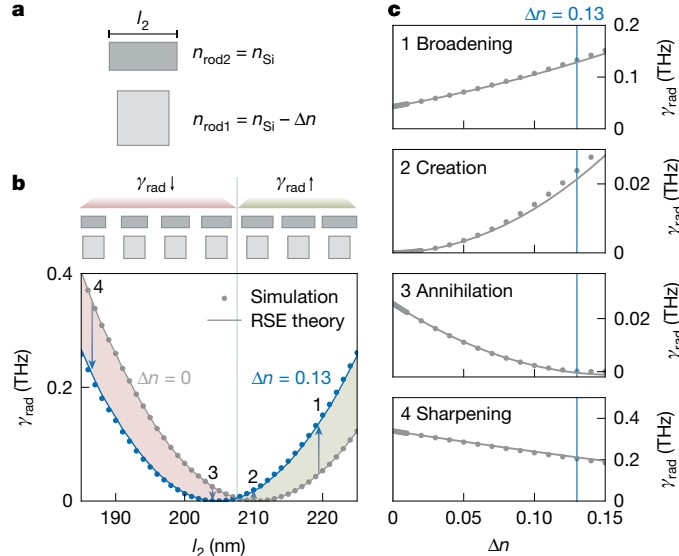

**Fig. 4 | RSE analysis of the change in refractive index near the RSP-BIC.**
**a**, Unit cell in which the refractive index of rod 1 ($n_{rod1}$) is reduced by $\Delta n$ with respect to the constant index of rod 2 ($n_{rod2}$). **b**, Simulation results (dots) compared with RSE theory (solid lines). $\gamma_{rad}$ is plotted versus $l_2$ for $\Delta n = 0$ (grey) and for the experimentally estimated $\Delta n = 0.13$ (blue). Increasing $\Delta n$ shifts the RSP-BIC to smaller $l_2$, so depending on $l_2$, moving from the grey to the blue curve either increases (green-shaded region) or decreases (red-shaded region) $\gamma_{rad}$. **c**, Four representative metasurfaces (1–4) and their corresponding $\gamma_{rad}$ are shown as a function of $\Delta n$ for: (1) mode broadening ($\gamma_{rad}$ increases), (2) dark-to-bright creation ($\gamma_{rad}$ increases from zero to a finite value), (3) annihilation ($\gamma_{rad}$ decreases approaching zero) and (4) sharpening of a bright mode ($\gamma_{rad}$ decreases).

vacuum. With these parameters, the RSP-BIC ($Q > 10^7$ in simulations) occurs at $l_2 = 210.5$ nm. Treating the pump-induced permittivity change as a perturbation, $\Delta\varepsilon = (n_0 - \Delta n)^2 - n_0^2 \approx -2n_0\Delta n$, the RSP-BIC, which has complex eigenfrequency $\omega_2$, can be hybridized following RSE theory with a parallel electric-dipole mode ($\omega_1$). Based on the hybrid eigenfrequency $\omega_{qBIC}$, an expression for $\gamma_{rad}$ can be found:

$$\gamma_{rad} = \text{Im}(\omega_{qBIC}) \approx \text{Im}\left(\omega_2\left[1 - v_2\Delta\varepsilon - \frac{u^2\omega_2(\Delta\varepsilon)^2}{\omega_1 - \omega_2}\right]\right). \quad (2)$$

The two perturbation matrix elements $v_2$ and $u$ arise from self- and cross-coupling of the RSP-BIC to the parallel-dipole mode, respectively (Supplementary Note 3).

Figure 4b tests the RSE by comparing $\gamma_{rad}$ from simulations (dots) with equation (2) (solid lines) as a function of $l_2$ for $\Delta n = 0$ (grey) and $\Delta n = 0.13$ (blue). These values are similar to the experimental results with and without the pump. The perturbation shifts $l_2^{RSP}$ to smaller values, creating two spectral regions. In the green-shaded region, $\gamma_{rad}$ increases when the pump is applied, whereas in the red-shaded region, $\gamma_{rad}$ decreases. Four representative cases are highlighted by arrows, and the corresponding $\Delta n$ sweeps are shown in Fig. 4c. Case 1 ($l_2 > l_2^{RSP}$) leads to an increase of $\gamma_{rad}$, broadening the mode. Case 2 ($l_2 = l_2^{RSP}$) creates a bright mode ($\gamma_{rad} > 0$) from a dark mode ($\gamma_{rad} \approx 0$). As the opposite of creation, case 3 ($l_2$ slightly smaller than $l_2^{RSP}$) yields a decrease of $\gamma_{rad}$ towards zero, annihilating the resonance at $\Delta n = 0.13$. Case 4 ($l_2 < l_2^{RSP}$) sharpens the mode by lowering $\gamma_{rad}$ while keeping it larger than zero. The close agreement of the simulations and RSE theory confirms the applicability of the derived analytical model for the given refractive-index perturbation range.

To experimentally explore these four switching cases, we perform pump–probe measurements at four corresponding positions along the $l_2$-gradient metasurface shown in Fig. 2f–i. Figure 5a is a sketch of the expected pump-induced spectral change. The pump creates

$\Delta n = 0.13$, shifting the resonance towards shorter wavelengths. Furthermore, the RSP-BIC condition shifts to smaller $l_2$ in accordance with Fig. 4b. The first case shown in Fig. 5b with $l_2 = 236$ nm and $p_1 < p_2$ features a visible SP-BIC mode around 799 nm. Upon pumping, the difference between the two dipole moments further increases. Hence, the mode broadens and the resonance amplitude as well as $\gamma_{rad}$ (from 0.04 to 0.14 THz) significantly increases before gradually returning within 20 ps. The second case at the RSP-BIC condition (Fig. 5c) with $l_2 = 226$ nm and $p_1 = p_2$, initially features no resonance. After pumping, the balance shifts to $p_1 < p_2$ and the resonance is created. The newly formed mode at 787 nm features $\gamma_{rad}$ of up to 0.058 THz, which decays back to values close to zero within 10 ps, when the RSP-BIC condition is restored. Note that for times below 0 ps and above 10 ps, the TCMT fit is not conclusive due to the absence of the mode. Supplementary Note 7 shows a corresponding power series, demonstrating continuous $\gamma_{rad}$ tunability. For the third case (Fig. 5d) with $l_2 = 210$ nm and $p_1 > p_2$, the pump pulse reduces the dipole imbalance, effectively restoring the symmetry with $p_1 = p_2$. Thus, the initial resonance is annihilated as $\gamma_{rad}$ drops from 0.025 to 0 THz. The resonance reappears after 20 ps as the system recovers. Finally, the fourth case, shown in Fig. 5e, with $l_2 = 186$ nm and $p_1$ much larger than $p_2$, the pump reduces $p_1$, but $p_1 > p_2$ still holds. Rather than restoring the symmetry, the dipole imbalance is reduced but remains non-zero. This causes a sharpening of the resonance with a decrease in amplitude, whereas $\gamma_{rad}$ drops from 0.35 to 0.12 THz, and the total $Q$ factor increases by 150% from around 100 to 250 (Supplementary Note 7). Note that the metasurface treated within the RSE (arrows in Fig. 4b) were selected to have initial $\gamma_{rad}$ values equal to those obtained experimentally in Fig. 5b–e at $\Delta n = 0$. Using the same $\Delta n = 0.13$, the changes of $\gamma_{rad}$ predicted by RSE closely match the pump–probe measurements for each case.

## Conclusion

In this work, we present an experimental demonstration of radiative-loss tuning in metasurfaces through temporal symmetry breaking, which allows us to couple or decouple a mode from the far field. Central to this innovation are RSP-BICs, which enable photonic symmetry in systems with broken geometric symmetry. This allows selective resonant pumping, which provides precise control over the asymmetry and radiative loss. We experimentally achieved resonance broadening ($\Delta Q = 100$, a change of 25%), sharpening ($\Delta Q = 150$, a change of 150%), and resonance creation and annihilation on 300-fs timescales, which is confirmed by the resonant state expansion model. We can continuously tune $\gamma_{rad}$ from 0 to 0.11 THz (divergent $Q_{rad}$ down to 3,300) by the pump fluence, with the estimated change of $\gamma_{int}$ being only approximately 14%. This underlines the high selectivity of our method.

The ability of $\gamma_{rad}$ to precisely control resonance-cavity parameters (for example, field enhancement and $Q$ factor) as well as the ability to toggle between a resonant and non-resonant system offers many possibilities throughout active nanophotonics, with several key examples like polaritonic strong coupling and ultrafast pulse modulation detailed in Supplementary Note 9. In addition to enabling low-loss, all-optical switching for telecommunications or computing, this direct control over radiative coupling offers significant advantages for time-resolved enhanced light–matter interactions, such as quantum emission and polariton-based effects. This is in contrast to conventional $\omega_0$ and $\gamma_{int}$ tuning methods, which struggle to dynamically control the local field enhancement or spectral width without introducing significant parasitic losses. Furthermore, the demonstrated rapid onset can induce significant time-varying dispersion[46]. This can further be used for time crystals[47], which so far have had to rely on the optical modulation of intrinsic resonances, like epsilon-near-zero modes, to achieve significant effects. Our technique is not limited to silicon-based systems and can be applied to various low-loss dielectrics that can be optically pumped, such as gallium arsenide[35]. Additionally, our method could be

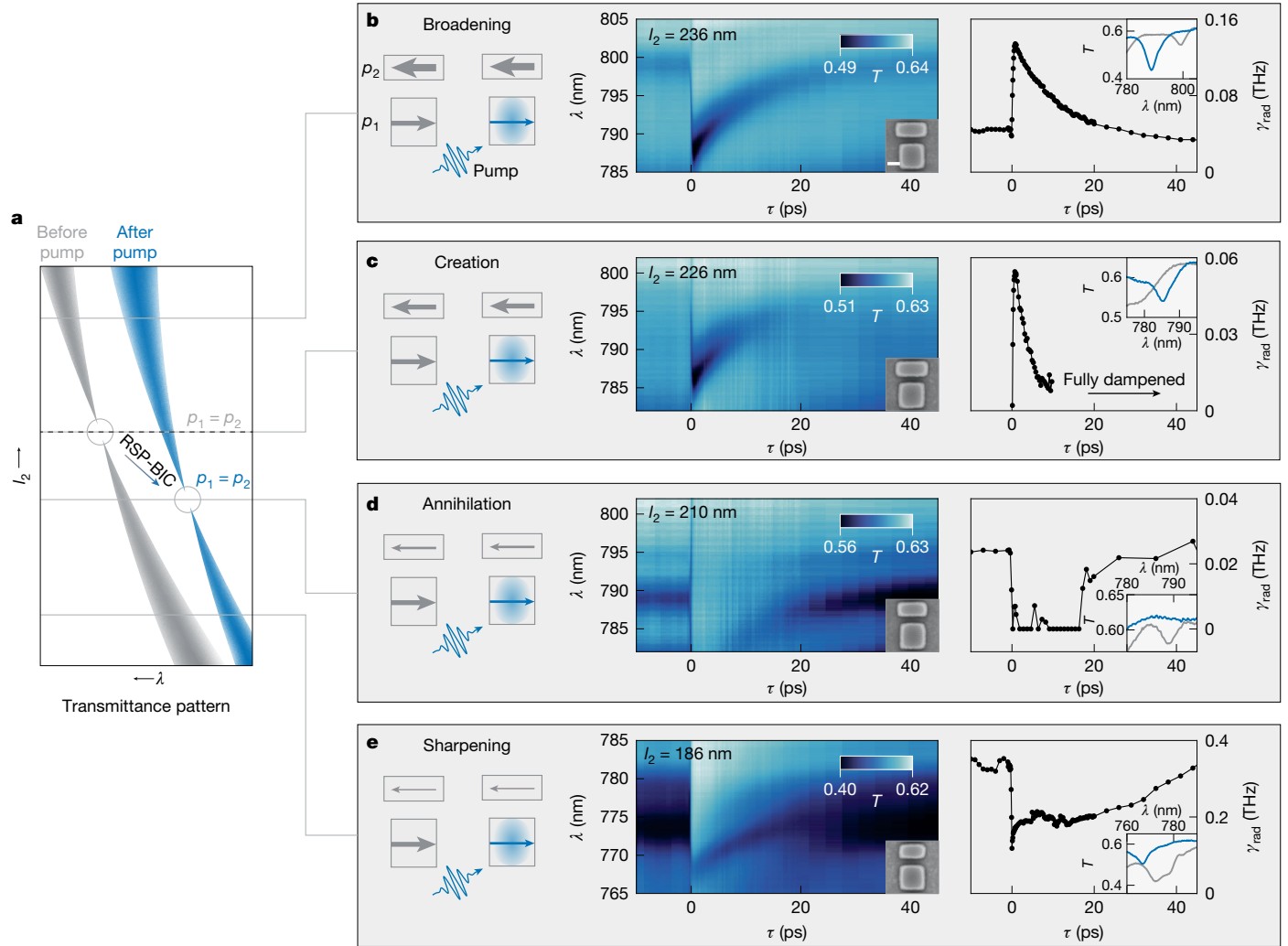

**Fig. 5 | Temporal radiative-loss tuning around the RSP-BIC condition.**
**a**, Illustration of the SP-BIC mode around the RSP-BIC condition for an $l_2$ sweep before (grey) and after (blue) pumping of the structure. **b**–**e**, Four key positions are highlighted with grey cuts where resonances are broadened (**b**), created (**c**), annihilated (**d**) or sharpened (**e**), all dependent on the initial ratio of $p_1$ to $p_2$. **b**, Transmittance time evolution with a 100 μJ cm$^{-2}$ pump pulse at 720 nm of the gradient position with $l_2 = 236$ nm, where $p_1 < p_2$, showing an increase in resonance amplitude and the radiative loss $\gamma_{rad}$. **c**, Time evolution with $l_2 = 226$ nm,

where $p_1 = p_2$ for the initial structure. After pumping, a mode is created, which quickly decays after 10 ps. This is also reflected in the $\gamma_{rad}$ fit, which increases from 0 to 0.058 THz before exponentially decaying. The fit is restricted to the time interval where the resonant mode is visible. **d**, Time evolution with $l_2 = 210$ nm, where $p_1 > p_2$, showing the annihilation of the BIC mode, which reappears after approximately 20 ps. **e**, Time evolution with $l_2 = 186$ nm, where $p_1 > p_2$, resulting in a decrease of $\gamma_{rad}$ and sharpening of the mode. Insets in **b**–**e** are the transmittance spectra at $t = 0$ and 1 ps. Scale bar, 100 nm (**b**).

extended to faster switching mechanisms based on nonlinear optical effects, such as the Kerr effect[48], enabling even shorter switching times and expanding its utility for ultrafast applications.

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

## Methods

### Simulations

We conducted the simulations using CST Studio Suite (Simulia), a commercial finite-element solver. The set-up included adaptive mesh refinement and periodic boundary conditions in the frequency domain. Crystalline silicon was modelled according to the data in ref. 49. For sapphire and $SiO_2$, we applied constant refractive indices of 1.75 and 1.44, respectively, assuming no material losses. The height of the $SiO_2$ layer was not considered. As this layer is significantly thicker than the silicon structure, the $SiO_2$/air interface was excluded from the model. For the RSE model, the eigenstate problem was solved using the Electromagnetic Waves Frequency Domain module of COMSOL Multiphysics in three-dimensional mode. A tetrahedral spatial mesh for the finite-element method was automatically generated by the physics-controlled preset in COMSOL. Simulations were performed within a rectangular spatial domain containing a single metasurface unit cell with periodic boundary conditions applied to its sides.

### Fabrication workflow

As basis for the sample, we used a commercially available 150-nm crystalline silicon film on a sapphire substrate (Si(100) Epi on R-Plane Sapphire, Roditi International Corporation Ltd). First, we etched the 150-nm silicon film down to 115 nm using inductively coupled plasma reactive-ion etching (ICP-RIE) in a $Cl_2$/Ar gas mixture (PlasmaPro 100 system, Oxford Instruments). Next, we deposited a 40-nm layer of amorphous chromium (Cr) through sputtering (AMOD, Angstrom Engineering Inc.). We spin-coated 125 nm of an electron-beam lithography resist (CSAR 62, Allresist GmbH) and wrote an inverse pattern of the two rods (eLINE Plus, Raith GmbH) at 30 kV with a 10-μm aperture. The patterned film was developed in an amyl acetate bath, followed by a bath of methyl isobutyl ketone and isopropyl alcohol (1:9 ratio). The Cr layer was then patterned by ICP-RIE using the resist as a mask and a $Cl_2$/$O_2$ gas mixture. The remaining resist was removed using Microposit Remover 1165 (Microresist GmbH). The patterned Cr was subsequently used as a hard mask for etching the silicon layer with ICP-RIE in a $Cl_2$/Ar gas mixture (PlasmaPro 100 system, Oxford Instruments). The Cr mask was removed with a chromium wet etchant (Chromium Etchant Standard, Merck KGaA). Finally, the sample was encapsulated by spin-coating undoped spin-on-glass (NDG-7000, Desert Silicon LLC) at 1,000 rpm, followed by baking at 150 °C for 30 min. The fabrication workflow is sketched in Extended Data Fig. 1.

### Gradient metasurface design

To achieve continuous spectral asymmetries, the gradient metasurfaces were manufactured with varying geometries, one being tuned with the width of the first rod, the second with the length of the second. The $w_1$ gradient metasurface varied from $w_1 = 95$ to 185 nm with the other parameters fixed as $p_x = p_y = 420$ nm, $l_1 = l_2 = 175$ nm and $w_2 = 95$ nm. The $l_2$ gradient metasurface varied from $l_2 = 175$ to 280 nm with fixed $p_x = p_y = 420$ nm, $l_1 = 175$ nm and $w_1 = 185$ nm. Both metasurfaces were periodic along the $y$ direction, whereas $w_1$ and $l_2$ changed smoothly along the $x$ direction, with an approximate step size of 0.4 nm between neighbouring unit cells. This is visible in Extended Data Fig. 2a, as the optical image shows a gradual colour change. SEM images (Extended Data Fig. 2b–d) visualize selected geometries. These different asymmetries lead to spatially varying transmittance spectra, as visualized in Extended Data Fig. 3 for the two gradients.

### Steady-state transmittance measurements

The steady-state transmittance measurements were performed using a confocal microscope (WITec Wissenschaftliche Instrumente und Technologie) with a white light source (Thorlabs OSL2 Fiber Illuminator) collimated with a collimator. A high-magnification objective (×100, numerical aperture = 0.9, Zeiss AG) collected the transmitted light, which was measured with a WITec microscope. To reduce the collection area to approximately 1 μm in diameter on the sample surface, an aperture was placed after the collecting objective. This small collection area was essential for reducing the spectral response to small regions within the spatially varying gradient metasurfaces. The entire gradient was then measured using a stepwise rastering of the sample in the $x$–$y$ plane to create a spectral map, from which we extracted the relevant data.

### Time-resolved spectroscopy

Time-resolved measurements were performed using a mode-locked Yb:KGW laser (Pharos Ultra II) at a 200-kHz repetition rate pumping an optical parametric amplifier (ORPHEUS-HP, Light Conversion) to generate laser pulses with a tunable wavelength of roughly 190-fs duration. As shown in Extended Data Fig. 5, the output was tuned to the wavelength of the pump mode and split into two using a beam splitter. One part (pump path) went directly to the sample, whereas the other (probe path) was focused onto a sapphire crystal to generate a supercontinuum. After passing a long-pass filter to filter out the pump wavelength and a delay stage to control the time delay between pump and probe, it was recombined with the pump path using a beam splitter. Both pump and probe polarizations could be controlled independently using half-wave plates. The pump and probe beams were condensed onto the structure using a ×10 objective with a 0.25 numerical aperture. Within the narrow spectral range investigated for the fabricated structures, the different probe arrival times for the spectral components due to chirp induced by the set-up were negligible.

To achieve large-area, uniform illumination, the pump path contained another lens for the back focal plane of the objective. This configuration led to only a minimal angular spread of the collected light, thus keeping the angular spread-induced broadening to a minimum (Supplementary Note 10). The transmitted supercontinuum was collected with another ×10 objective with a 0.25 numerical aperture and analysed with a spectrometer (Princeton Instruments Acton SP2300) using a silicon CCD (Princeton Instruments Pixis 100 f) and a 300 g mm$^{-1}$ grating. The transmitted pump was filtered out using another long-pass filter in front of the spectrometer.

### Data availability

The data that support the findings of this study are available at Zenodo (https://doi.org/10.5281/zenodo.15662526)[50].

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

**Acknowledgements** This project was funded by the Deutsche Forschungsgemeinschaft (German Research Foundation) under grant numbers EXC 2089/1–390776260 (Germany's Excellence Strategy) and TI 1063/1 (Emmy Noether Program), the Bavarian programme Solar Energies Go Hybrid (SolTech), the Enabling Quantum Communication and Imaging Applications project (EQAP), and the Center for NanoScience at LMU. It was also funded by the European Union (ERC, METANEXT, 101078018, and EIC, OMICSENS, 101129734). The views and opinions expressed are, however, those of the authors only and do not necessarily reflect those of the European Union, the European Research Council Executive Agency, or the European Innovation Council and SMEs Executive Agency. S.A.M. additionally acknowledges the Lee-Lucas Chair in Physics. We are grateful to M. Gorkunov and B. Tilmann for many useful discussions.

**Author contributions** A.A. and A.T. conceived the idea and planned the research. A.A. fabricated samples and performed the linear measurements. T.P. performed the time-resolved measurements. A.A., T.W. and A.A.A. conducted the numerical simulations. A.A., T.P. and

T.W. contributed to the data processing. A.A., T.P., A.A.A., T.W. and A.T. contributed to the data analysis. A.A.A. derived the RSE model. S.A.M., L.d.S.M. and A.T. supervised the project. All authors contributed to the writing of the paper.

**Funding** Open access funding provided by Ludwig-Maximilians-Universität München.

**Competing interests** The authors declare no competing interests.

**Additional information**
**Correspondence and requests for materials** should be addressed to Stefan A. Maier or Andreas Tittl.

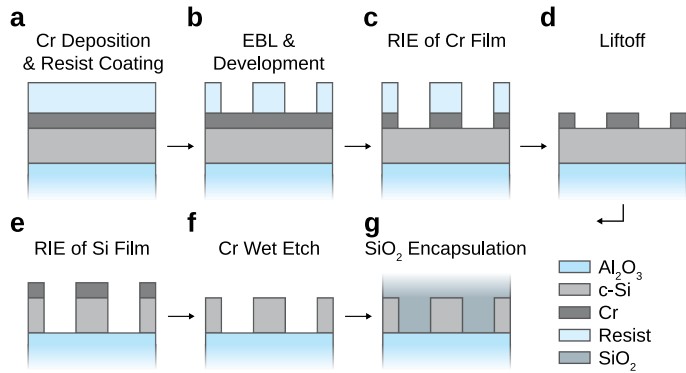

**a** Cr Deposition & Resist Coating
**b** EBL & Development
**c** RIE of Cr Film
**d** Liftoff

**e** RIE of Si Film
**f** Cr Wet Etch
**g** SiO$_2$ Encapsulation

Al$_2$O$_3$
c-Si
Cr
Resist
SiO$_2$

**Extended Data Fig. 1 | Fabrication Workflow. (a)** The 115 nm crystalline silicon (c-Si) film on Al$_2$O$_3$ is first coated with a 40 nm layer of Cr via sputtering, followed by spin-coating with 125 nm of electron beam lithography (EBL) resist. **(b)** The resist is exposed in an EBL machine and developed using a two-step process. **(c)** The patterned resist serves as a mask for anisotropic etching of the Cr film using reactive ion etching (RIE). **(d)** During the liftoff the remaining resist is chemically removed. **(e)** The Cr layer is then used as a hard mask to etch down the 115 nm c-Si film. **(f)** The remaining Cr is chemically wet-etched. **(g)** Finally, the c-Si film is encapsulated in spin-on-glass (resembling SiO$_2$) with approximately 1000 nm height.

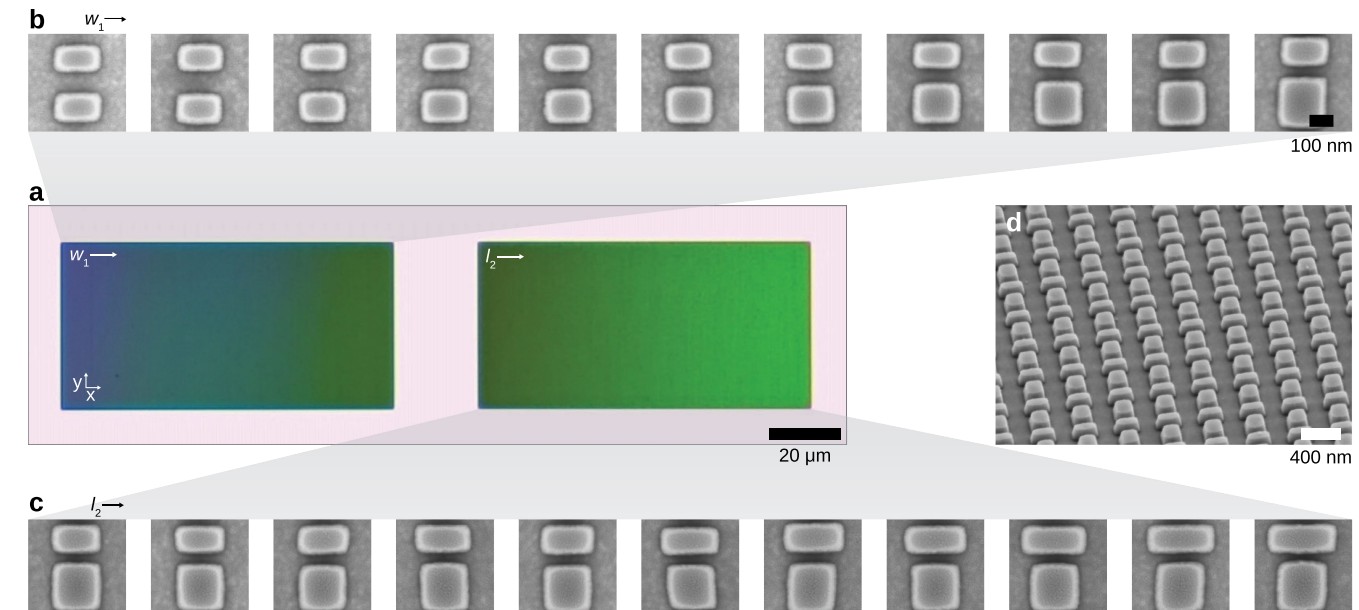

**Extended Data Fig. 2 | Gradient Metasurfaces for Continuous Tuning of $\gamma_{rad}$.**
(**a**) Single optical image of the substrate containing side by side the $w_1$ gradient (left) and the $l_2$ gradient metasurface (right). (**b**) SEM images of individual unit cells taken at 10 µm intervals along the x-axis for the $w_1$ gradient, starting from 0 to 100 µm. (**c**) Equivalent SEM images along the $l_2$ gradient. (**d**) Angled SEM image of the $l_2$ gradient around the RSP-BIC condition.

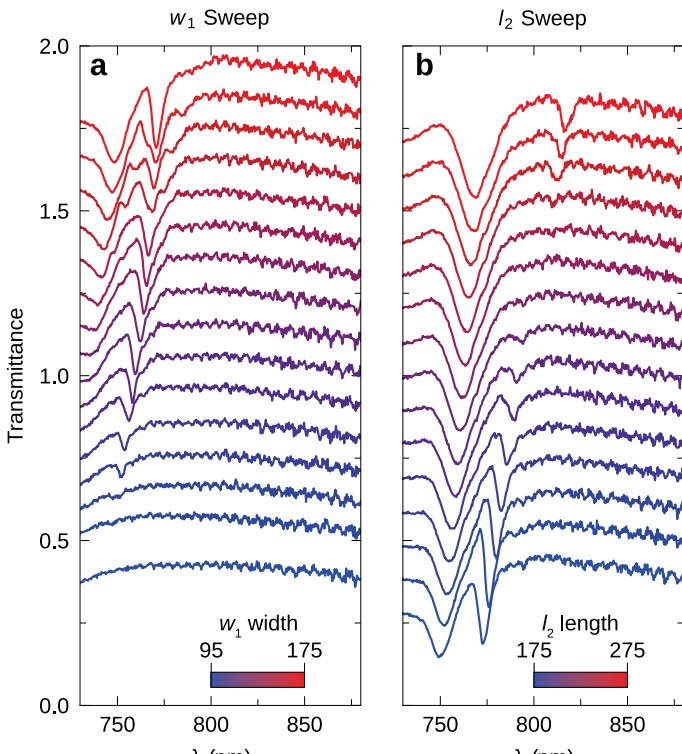

**Extended Data Fig. 3 | Transmittance Spectra for $w_1$ and $l_2$ Gradient Metasurfaces.** (**a**) Transmittance spectrum for varying $w_1$ (equivalent to Fig. 2h, left) indicated by the blue-to-red color map. The transmittance is offset by 0.1 for clarity. (**b**) Transmittance spectrum for varying $l_2$ (equivalent to Fig. 2h, right), plotted similarly to (a). The white light used for taking the spectra was polarized along the x-axis of the sample.

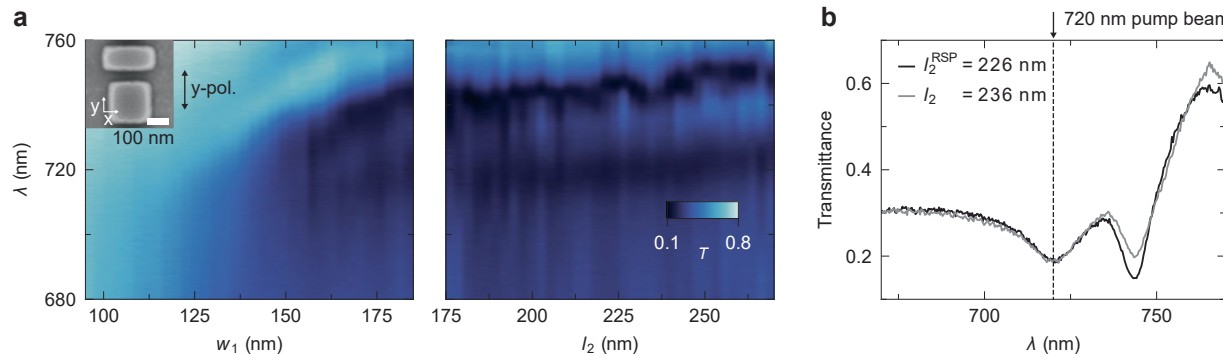

**a**

**b**

**Extended Data Fig. 4 | Y-Polarized Transmittance Spectra for $w_1$ and $l_2$ Sweeps. (a)** Transmittance spectra for y-polarized light (inset for reference) in analogy to the x-polarized spectra of the two gradient metasurfaces in Fig. 2h: on the left for varying $w_1$ with fixed $l_1 = l_2 = 175$ nm and $w_2 = 95$ nm; on the right for varying $l_2$ with fixed $w_1 = 185$ nm, $w_2 = 95$ nm, and $l_1 = 175$ nm. In the $w_1$ sweep (left), the Mie 1 and Mie 2 modes discussed in Fig. 3 are not visible for small $w_1$ and appear as $w_1$ increases. In the $l_2$ sweep (right), there is almost no change in the transmittance spectra, indicating the mode's independence from the size parameters along the off-polarization axis. This allows us to use the same optimized pump wavelength of 720 nm, as it remains resonant with Mie 1 across the entire $l_2$ gradient. This is confirmed in (**b**) by the transmittance spectra at the RSP-BIC condition with $l_2^{RSP} = 226$ nm (shown in Fig. 3a) and with $l_2 = 236$ nm (the structure used in Fig. 3e,f), which both feature almost identical Mie 1 modes.

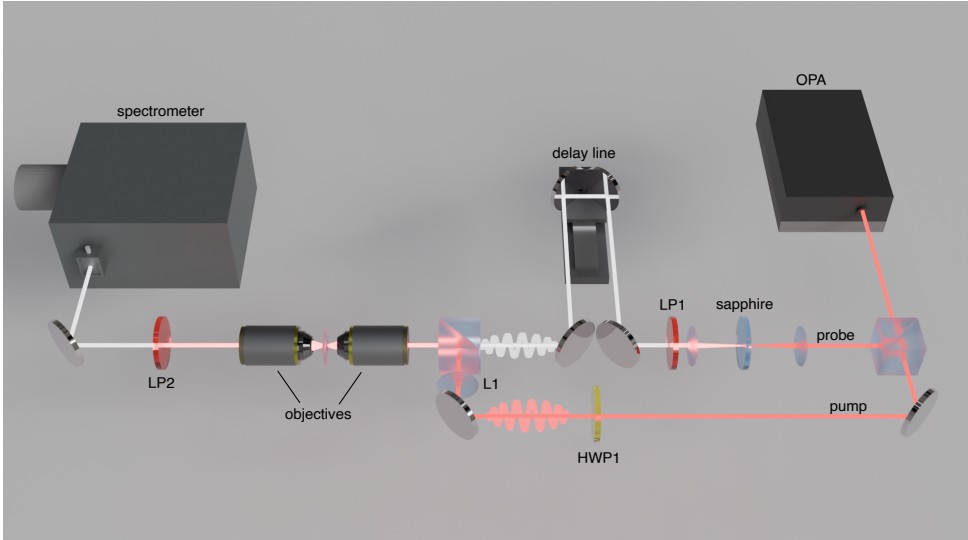

**Extended Data Fig. 5 | Schematic of the Experimental Setup.** Pump-probe setup including an optical parametric amplifier (OPA), two 750 nm long pass filters (LP1 and LP2), a lens L1 and a broadband half-wave plate HWP1.

**a  Non Resonant Pumping of SP-BIC**

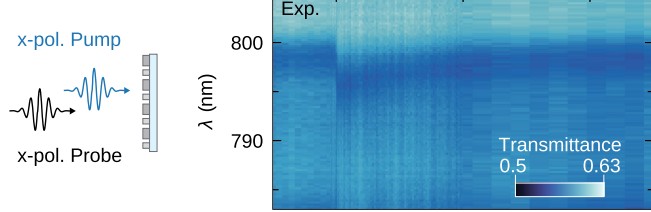

**b  Mie 1 Resonant Pumping of SP-BIC**

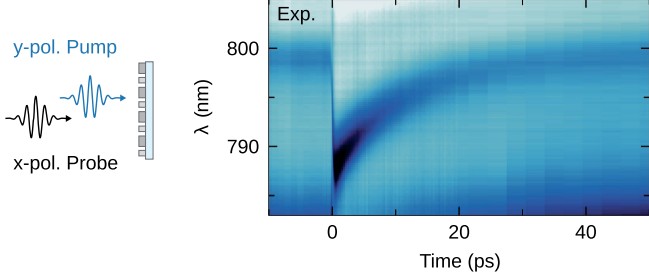

**Extended Data Fig. 6 | Non resonant pumping vs. resonant pumping.**
(**a**) Pump-probe spectral time trace (720 nm pump with a fluence of 100 µJ/cm²)
for a metasurface at the resonance broadening position, where both pump
and probe pulses are x-polarized. The SP-BIC at 799 nm shifts by 3.9 nm and
exponentially returns to the initial wavelength with a time constant of 20 ps,
with minimal changes in resonance amplitude. (**b**) Same configuration as in (**a**)
but with the pump pulses y-polarized, resonantly pumping Mie 1. The SP-BIC
shifts 9.2 nm, and increase in resonance amplitude suggests an increase in $\gamma_{rad}$
due to the experiments being performed in the undercoupled regime.