## [Peer Review File · Nature]

Optical Control of Resonances in Temporally Symmetry-Broken Metasurfaces

Corresponding Author: Professor Andreas Tittl

Version 0:

Reviewer comments:

Referee #1

(Remarks to the Author)

In their manuscript, Aigner et al. experimentally realize ultrafast control of optical resonances in qBIC-based metasurface through optically induced symmetry breaking. Overall, I find the paper very well-written and the results very interesting and appealing to the field of metasurfaces and ultrafast optics. However, in my opinion the paper lacks the outstanding novelty that would warrant a publication in Nature. In fact, the idea of controlling the radiative loss through optically induced “symmetrization” of qBIC structure is not new; to the best of my knowledge, it was introduced for the first time in the work by Yang et al. (<https://doi.org/10.1515/nanoph-2023-0718>). While I am not aware of any experimental realizations of this effect so far, it certainly limits the novelty of the present research and at least should be mentioned in the state-of-the-art. Also, in my opinion the authors don't provide a convincing case for the broader impact of their results, which suggests that the paper might be better suited for a more specialized journal. Therefore, I am afraid I cannot recommend this manuscript for publication in Nature. At the same time, I have a list of questions and comments that could help further improve the paper before publication elsewhere.

The main point that concerns me is the fact that the authors focused on the modifications of the radiative losses while almost fully neglecting the non-radiative part. This sublimates into the following questions:

(i) The authors use TCMT to extract the radiative losses from the spectroscopy data. However, the reliability of this extraction relies heavily on the accuracy of the measured peak amplitude. In a realistic scenario, the measured transmission spectra would represent an average over a finite range of angles of incidence. Depending on the dispersion of the modes of the structure, this can lead to considerable broadening and suppression of the resonance. As I did not find a very detailed description of the measurement configuration, I could only assume that the spectra is measured with the supercontinuum probe covering the full NA of the objective (0.25). The authors should consider how this influences the accuracy of their estimations on γ_{rad} (and, by extension, γ_{int}).

(ii) The authors claim that the optical pumping they provide changes the radiative losses, while “the effect on γ_{int} is negligible” (page 3, line 89). This needs a more detailed discussion. In fact, the photogenerated free carriers contribute significantly to both the real and imaginary parts of the refractive index. In the short wavelength range, the changes to n and k could be of comparable magnitude. The observed spectral shifts of the resonance (up to 10 nm) indicate quite strong change of n , which implies considerable change in k . Figure S11 reveals that the extracted non-radiative losses are almost 20x larger than the radiative part - what authors probably imply is that the changes to γ_{int} are small with respect to its original value; this needs to be detailed earlier. It is particularly important in the context of the outlook that the authors provide in the conclusion (...This is in contrast to conventional ω_0 and γ_{int} tuning methods, which struggle to dynamically control local field enhancement or spectral width without introducing significant parasitic losses...). Both the local field enhancement and linewidth are defined by the balance of radiative and non-radiative losses, and with the initial strong imbalance of the two (at least, based on the measured data) I don't see an obvious benefit of controlling the radiative part.

(iii) The apparent uncertainty in the extracted γ_{rad} and γ_{int} is particularly important in the context of the discussion of data in Figs. 4 and 5. The authors attribute the change of refractive index solely to the contribution of

photogenerated free carriers. If that is the case, the relaxation time of the real and imaginary part of the refractive index should be similar. Instead, what the authors observe is a fast decay of γ_{int} that they attribute (quite vaguely) to nondegenerate photon absorption and to the cavity resonance lifetime. However, why does the longer cavity lifetime contribute to the non-radiative but not radiative losses?

(iv) To my opinion, the paper will benefit a lot from the estimations of the induced refractive index changes and a connected discussion of the limitations of the approach for creation/annihilation of more pronounced optical resonances.

Some minor technical points are listed below:

- (a) The authors should use the absolute values of transmittance in the colorbar of Fig. 3e,f.
- (b) The Supplementary Materials would benefit from short descriptions of the figures that go beyond the captions to make it more autonomous.
- (c) Can the authors comment on the inconsistency of the transmittance amplitude measured in linear spectra (Fig. 1h) and time-resolved spectra? This is likely connected with the major point (i) above.
- (d) The discrepancy of spectral profile of the modes in Fig. 2d and h is also too obvious to completely neglect in the discussion.
- (e) Fig. 3e,f seem to indicate different incident angles for pump and probe. Does this correspond to any realistic geometry? Taking into account the point (i) above, maybe this schematic could be improved.
- (f) The part of discussion related to Sb₂S₃ seems too unrelated to the core results of the paper. The utilization of phase-change materials, especially antimony compounds, has its own challenges which deserve a separate research effort. In the current state, I feel that it breaks the otherwise very smooth flow of the paper.
- (g) The “femtosecond timescale” of creation and annihilation mentioned by the authors is a bit misleading, I would suggest using “sub-picosecond” instead - this does not diminish the quality and significance of the results anyways.

Referee #2

(Remarks to the Author)

This manuscript presents a first-of-its-kind experimental demonstration of a resonant optical structure with an ultrafast tunable radiative decay rate of the resonant mode. The manuscript presents a highly original and elegant idea supported with a thorough experimental verification. Overall, this is a very well-executed, well written and structured manuscript which I enjoyed reading as a referee. I particularly want to emphasize the excellent presentation; the manuscript is easy to follow. That being said, I do have a number of serious concerns about the interpretation and the analysis of the experimental data, which need to be thoroughly addressed before the manuscript can go any further down the publication chain.

1. I believe the sketch in Fig. 1(d) could be more informative if the authors show not only the annihilation of the resonance, but also illustrate its progressive narrowing/broadening, which is the key result of γ_{rad} variation.
2. In caption of Fig. 1(f) did the authors imply “resulting in asymmetric dipole moments and the emergence of the quasi SP-BIC mode”?
3. Does the condition $p_{\text{tot}} = 0$ in an asymmetric metasurface guarantee an exact BIC with infinite lifetime, or this is just a good approximation? The SP-BIC is an exact solution of Maxwell’s equations with exactly zero radiative loss. Does that also work with the RSP-BIC? Although the total dipole moment of the unit cell is zero, I can imagine its total higher-order moments (clearly being non-zero due to the geometric asymmetry) producing a $k_x = 0$ Bloch mode with non-zero radiation.
4. What exactly is the “loss density distribution” that the authors present in Fig. 3? This is the first time I’m encountering such a terminology in nanophotonics.
5. One of my key criticism relates to the results presented in Fig. 3(f) and their interpretation in the text. The authors’ interpretation of these measured spectra is “... the increase in resonance amplitude suggests a change in γ_{rad} ”. A change of γ_{rad} (in a nearly lossless system) would result in a narrowing/broadening of the resonant peak/dip/Fano resonance (depending on the background scattering matrix of the system), but its “amplitude” would remain unchanged. Such a behavior easily follows from the single-mode CMT model that the authors employ throughout their study. In Fig. 3f I do not see any such narrowing or broadening of the resonant line, but instead I do see a substantial change in the amplitude of the transmission dip, which cannot happen solely as a result of γ_{rad} variation in a lossless single-mode system.
6. In that regard, I am very curious what the CMT fits of the measured transmission spectra look like. The main text as well as SI shows numerous plots of γ_{rad} vs time, but not a single CMT fit of the experimental data, which would be crucial to judge whether optical pump is indeed capable of modifying its purely radiative decay rate. Such fits would be very helpful for interpreting the data shown in Figs. 2(h) and (i), for example, where the transition from measured spectra to γ_{rad} occurs without showing explicitly any intermediate fitting step. Same about Figs. 4(b)-(e).
7. There is an argument to be made that, instead of considering the unit cell as a single entity with some “black box” CMT parameters, one could consider same exact metasurface unit cell as a combination of two separate particles supporting their own Mie-like resonant states, which couple in the near field by the dipole-dipole interaction. From that perspective, the effect of optical pump manifests as lowering S_1 refractive index, which in turn leads to the familiar spectral resonance shift of one of the two nanoparticles in the dimer. Which is exactly what the authors want to distance themselves by considering the dimer as a single black box with some resonant characteristics. That is to say, whether this work demonstrates tuning of the resonant frequency or the radiative decay rate depends on the perspective we analyze the system from.
8. Yet another piece of information that this work crucially lacks is any theoretical description of the refractive index modulation by an optical pump. While the authors do not focus on the refractive index-based description of the resonance

modification and prefer to describe its evolution in terms of CMT decay rates, providing even a simplest model of refractive index modulation would be helpful in two ways: (i) it would highlight the key mechanism enabling index modulation, and (ii) it could perhaps be valuable in analyzing the transient dynamics of the resonant characteristics shown in Fig. 5. Perhaps something like two-photon absorption + a bi-exponential charge carriers density decay would be sufficient for such a simple model.

9. If I'm not mistaken, Fig. 5 is the first instance where γ_{int} comes into the analysis of the optical system response for the first time. Quite surprisingly, the authors' analysis shows that γ_{int} does increase substantially during pumping. This could actually address the important point I was making in Comment 5. But if it does, the analysis performed for the data in Figs. 3 and 4 should then incorporate this γ_{int} (which at the moment it does not, I believe).

10. Page numbers in many of the references are either missing, or incorrect (the ones starting with page "1" for sure). Many of the references even miss journal names.

To conclude, this is an interesting, highly original, innovative and significant work for the field of nanophotonics and electromagnetism in general, which could be considered for publication in Nature if the authors do accurately address all the issues related to the analysis and interpretation of the experimental data.

Referee #3

(Remarks to the Author)

I co-reviewed this manuscript with one of the reviewers who provided the listed reports.

Referee #4

(Remarks to the Author)

In this work, Aigner, Possmayer and co-authors demonstrate, for the first time to my knowledge, the ultra-fast tuning of the radiative loss of a metasurface system. Compared to previous works, the main novelty lies precisely in that they deal primarily with the radiative part of the losses, by which they can create, annihilate and modulate the width of a resonance, namely a (quasi-)bound state in the continuum (BIC). Indeed, previous works that demonstrated tuning of similar type of modes in a metasurface dealt mainly with modulations of either its spectral position, its non-radiative (absorption) losses, or both. This radiative-loss tuning is here achieved using a clever idea. To illustrate it, first, the authors break the geometrical symmetry of the unit cell without breaking the optical symmetry, forming what they refer to a restored symmetry protected BIC (RSP-BIC). This mode exhibits a full cancellation of the net dipole moment despite being formed by two dissimilar nanorods. Interestingly, this opens the possibility of selectively pumping one of these rods, therefore being able to modulate its dipole moment. This is used to turn this RSP-BIC into a qBIC that can couple to external radiation (creation mode). The same concept is then used to turn into a (true) RSP-BIC a mode that was initially a qBIC one (annihilation mode). Partial states that induce sharpening and broadening of the resonance can also be accessed with slightly modified conditions.

While I have no doubts about the novelty of the approach, which is also creative and (as the authors point out) can be extended to other scenarios, I have more doubts about its significance. In particular, the authors claim that their results open "new possibilities for studying light-matter coupling effects[...]. This is in contrast to conventional ω_0 and γ_{int} tuning methods, which struggle to dynamically control local field enhancement or spectral width without introducing significant parasitic losses". In my opinion, however, the authors fail to show evidence for any of such scenarios. Can they show/illustrate one case in which this method yields effects not attainable by either ω_0 and/or γ_{int} tuning?

As shown in Fig. 4, all the effects shown (creation, annihilation, broadening and sharpening) have also some associated (temporal) spectral shifts. Intuitively, I would think that this is because these effects are still a consequence of a spectral shift, the only difference being that, in this case, the spectral shift happens to the modes of only one of the rods forming the unit cell. Is this the case? If so, what is the Δn that one needs to consider to explain the effects?

Version 1:

Reviewer comments:

Referee #1

(Remarks to the Author)

I greatly appreciate the outstanding effort the authors put in the revision. The revised version is better positioned against state-of-the-art, and the arguments the authors provide regarding the advantages of their approach over structured illumination are valid and convincing. This changes the (at least, my subjective) perspective on the novelty of the paper. The new supplementary section covering the possible applications is a very sound contribution to the impact of the paper, and other raised concerns were addressed in full. I therefore support the publication of the revised paper in Nature, with a few minor points mentioned below to be taken care of.

-I recommend still explicitly mentioning the original imbalance of radiative and internal losses of the system (their absolute values, as in Fig. S16) in the main text when discussing Fig. 3f - the percentage change is indeed insightful, but does not give the full understanding of the dynamics.

-Maybe the authors could briefly mention if they expect a considerable chirp for the probe pulse as it passes through multiple dispersive elements in the experimental setup and whether it could contribute to the extracted temporal dynamics for different resonance manipulation scenarios.

-I advise the authors to do a careful proofreading of SI, as there are several typos and errors in equations (in particular, line 452).

Referee #2

(Remarks to the Author)

I would like to thank the authors for their accurate revision and the detailed response to the referees' comments.

The authors have done extensive changes to the manuscript. Particularly, CMT fits of the measured spectra presented now in SI indeed confirm the reported behavior of radiative and non-radiative decay rates. I have no further technical comments and can recommend this manuscript for publication.

Referee #3

(Remarks to the Author)

I co-reviewed this manuscript with one of the reviewers who provided the listed reports.

Referee #4

(Remarks to the Author)

The authors have made a great effort in replying to all the questions and concerns raised by all the referees. I value the extra work that they have put on their reply, both adding new technical portions that helped clarified the interpretation of the observed effects, as well as new insights that help highlighting the significance of their work. In my opinion, the work can now be published in the journal.

Comments to the Referees

Referee #1

Reviewer comment:

In their manuscript, Aigner et al. experimentally realize ultrafast control of optical resonances in qBIC-based metasurface through optically induced symmetry breaking. Overall, I find the paper very well-written and the results very interesting and appealing to the field of metasurfaces and ultrafast optics. However, in my opinion the paper lacks the outstanding novelty that would warrant a publication in Nature. In fact, the idea of controlling the radiative loss through optically induced “symmetrization” of qBIC structure is not new; to the best of my knowledge, it was introduced for the first time in the work by Yang et al. (<https://doi.org/10.1515/nanoph-2023-0718>). While I am not aware of any experimental realizations of this effect so far, it certainly limits the novelty of the present research and at least should be mentioned in the state-of-the-art.

Answer:

We thank the Reviewer for their positive evaluation, for acknowledging that our results are interesting and appealing, and for complimenting our writing style. Indeed, Yang et al. show an interesting method on how interference patterns can be used to boost the Q-factor of metasurfaces based on simulation results. We also sincerely apologize for not mentioning and citing the work of Yang et al. in our initial submission. We have now added this important reference in the revised version.

Furthermore, in the paragraphs below, we present detailed arguments on why our work presents a conceptually and experimentally distinct approach from Yang et al. and how it provides the first realization of complete radiative-loss control in metasurfaces.

1. Conceptual Distinction: Bidirectional Control over the Radiative Q-factor

While Yang et al. offer an interesting numerical proposal to change the Q-factor by reducing radiative losses, they do not investigate resonance broadening or creation in their geometry. Instead, the focus lies on resonance sharpening and annihilation, neglecting a key result of our work.

2. Experimental Feasibility: Unstructured Pump Beams vs. Interferometric Multibeam Patterns

Yang et al. propose creating an interference fringe pattern at the sample by crossing two beams at carefully chosen angles. In principle, the goal is that the pattern selectively excites one resonator per unit cell. However, in practice this is highly challenging due to three reasons:

- **Pitch Matching Over Large Arrays:** A typical metasurface supporting collective BIC modes often spans tens or hundreds of unit cells in each lateral direction. As one example, we can consider an array of 50×50 unit cells with a unit-cell pitch of ~ 550 nm (the geometry proposed by Yang et al.). For the interference pattern to pump only one resonator (and not its neighbor), the fringe spacing must exactly match the 550 nm pitch. Even a 1 nm mismatch in fringe spacing over 50 unit cells compounds to 50 nm of relative shift, producing unwanted lateral shifts of the intensity peaks and valleys, leading to non-uniform refractive index changes over the size of the array.
- **Gaussian Beam Curvature and Limited Collimation:** The two interfering pump beams are typically Gaussian and not plane waves. Even if one assumes identical angles for both beams, the finite beam waist means only the very center region can maintain near-constant phase fronts and near-constant fringe spacing. Outside that central region, the beams begin to converge or diverge, slightly altering the fringe spacing. Over tens of micrometers (the size of a typical metasurface region), this shift can accumulate enough misalignment to prevent the selective pumping effect.
- **Energy and Complexity Costs:** Even if one attempts the use of extremely large beam diameters to reduce curvature, the required fluence to reliably form perfect fringes of sufficient intensity over the entire area would be highly energy-inefficient and experimentally challenging. An additional delay line is needed to ensure consistent femtosecond overlap of the two probe pulses as well as multiple high-precision optical components for beam steering. Further, such an approach would not allow for collinear pump-probe beams (i.e., the individual beams need to follow different paths until they meet at the sample position).
- **Temporal and spatial stability:** In addition to matching the interference pattern with the grating's pitch, the fringes also need to be aligned properly. A lateral translation of the sample of 225 nm will result in pumping the wrong part of the unit cell, as will a change in the path length difference between the two pulses of 266 nm due to the opposite phase at the point of interference.

In sharp contrast, each unit cell of our design contains two resonators with different Mie modes, so an unshaped, single laser pulse naturally resonates in only one of the two resonators. This fundamental spatio-spectral selectivity of our approach replaces the need for delicate alignment of spatial interference patterns. This advantage is what makes our resonance creation and annihilation experiments possible with robust performance, where the approach by Yang et al. would most likely be constrained and impeded by experimental limitations.

3. Fundamental tradeoff between array size and switching speed

To achieve interference fringes over a large area, the beams need to be as monochromatic as possible, limiting their minimum pulse duration. Assuming a coherence length $L = \frac{c}{\pi\Delta\nu}$ with speed of light c and linewidth $\Delta\nu$ limits the maximum path delay between pump and probe. To illuminate 50 unit cells in the proposed geometry, a lower limit of about 44 fs therefore arises for Fourier-limited sech^2 pump pulses, while already suffering from continuously decreasing modulation with distance from the center of the surface.

Furthermore, with ultrashort pulses, geometric problems start appearing at the proposed angles of incidence and array size: As the pump pulse arrives at an angle θ , one of its sides arrives at the sample earlier than the other, pumping the metasurface at different times. As the second pump pulse arrives with the opposite angle, the effective area of the interference pattern (where both wavefronts arrive at a similar time) would be given by $\frac{ct}{2\sin(\theta)}$ with the pulse length t , speed of light c and interference angle θ . To illuminate 50 unit cells in the proposed geometry, the lower limit in pulse duration by this effect is 44 fs again, while also suffering from less modulation contrast at the edges of the structure. Counteracting this effect through tilted wavefronts adds additional experimental complexity.

In contrast, the minimum pump pulse duration in our structure is only limited by the ~50 nm broad resonance of Mie 1 and 2, theoretically enabling pump pulses as short as 11 fs, without fundamental constraints in the size of the metasurface that can uniformly be illuminated.

4. Comparison with Existing Literature

Finally, we note that there have been several earlier attempts to modulate γ_{rad} mostly based on plasmonic metasurfaces. Such systems often exploit changes in free-electron concentration in metals (e.g., gold or indium tin oxide, where the plasma frequency can be modulated) to alter how strong the plasmonic mode radiates. However, only partial radiative tuning can be achieved with these methods because no complete on/off switching is possible. Thus, the novelty of our approach does not rely on tuning γ_{rad} per se but rather on coupling or decoupling a mode from the far field on ultrafast timescales using γ_{rad} tuning to the fullest extent.

In summary, in our manuscript we show that our approach is based on a fundamentally new mechanism based on three main points: (i) single-beam spectral selectivity instead of complex spatial interference, (ii) a true ability to both couple and decouple a mode from the far-field, and (iii) our successful experimental demonstration which was only made possible by implementing point (i). We hope these differences address the Reviewer's concern regarding novelty.

Action taken:

In the introduction on page 3 we added a sentence regarding Yang et al. featuring their work, and sharpening a distinction between their results and the aim of our manuscript:

While a recent numerical study investigates Q-factor boosting through interference-based pumping,¹ the ability to arbitrarily tune radiative loss to increase or decrease the Q-factor and achieve on/off switching has so far not been realized.

Reviewer comment:

Also, in my opinion the authors don't provide a convincing case for the broader impact of their results, which suggests that the paper might be better suited for a more specialized journal. Therefore, I am afraid I cannot recommend this manuscript for publication in Nature. At the same time, I have a list of questions and comments that could help further improve the paper before publication elsewhere.

Answer:

We thank the Reviewer for their feedback regarding the broader impact of our work. In response, we have now expanded the manuscript to include a section on the broader impact in the introduction and an additional paragraph in the conclusion. Furthermore, we added a supplementary section discussing multiple promising application examples throughout optics in greater detail.

Action taken:

We added the following sentences in the introduction on pages 2 and 3 to show, besides the novelty of γ_{rad} tuning, its potential broad impact on active photonics:

This capability is transformative for active photonics, particularly in applications like optical communications, signal processing, or filtering, where it can minimize spectral crosstalk and avoids parasitic losses. Furthermore, in contrast to ω_0 and γ_{int} , γ_{rad} directly governs the resonance amplitude, Q-factor, and local field enhancement, making it critical for precise control of metasurface functionalities.

...

Thus, the ability to arbitrarily tune radiative loss to increase or decrease the Q-factor and achieve on/off switching of resonances has so far not been achieved, impeding future applications of metasurfaces like telecommunication switches and optical filters.

We added the following section in the discussion on page 11:

In addition to enabling low-loss, all-optical switching for telecommunications or computing, this direct control over radiative coupling offers significant advantages for time-resolved enhanced light-matter interactions such as quantum emission and polariton-based effects.

Further, we added a more detailed discussion on various possible applications demonstrating the broad impact of our approach in a new supplementary note linked on page 11:

The ability of γ_{rad} to precisely control resonance cavity parameters (e.g., field enhancement and Q-factor) as well as being able to toggle between a resonant and nonresonant system offers many possibilities throughout active nanophotonics, with several key examples detailed in **Supplementary Note 12**.

Further, we added Supplementary Note 12

Supplementary Note 12: Potential applications for radiative loss based active photonics

The possible applications for γ_{rad} tuning are diverse. Below, we specifically name six possible directions where γ_{rad} tuning can provide improved performance and new functionalities compared with conventional approaches based on tuning ω_0 or γ_{int} .

1. On/Off Switchable Filters: Conventional approaches to resonance tuning, i.e., shifting ω_0 or increasing γ_{int} , cannot fully switch a resonance on or off. ω_0 tuning moves the mode away from its original spectral position, but it cannot create a completely transparent metasurface system because a highly reflective resonant mode is always present. γ_{int} tuning increases intrinsic losses and thus quenches the reflectance amplitude of resonance but still leaves some broadened residual mode, as the structure remains radiatively coupled, just with smaller amplitude and broader linewidth. Further, it typically leads to overall lower transmission. In contrast, γ_{rad} tuning can entirely decouple the mode from the far field using a system that transitions from truly “resonance-free” ($\gamma_{\text{rad}} = 0$) to fully “resonant” ($\gamma_{\text{rad}} > 0$). The result is an on/off switchable filter that, in its “off” state, is truly transparent, minimizing spectral crosstalk and absorption. This capability is important for advanced active optical filtering in photonic circuits.

2. On-Demand Sensors: In sensing applications, the ability to activate or deactivate a resonance at will can help to keep an optical system transparent and only switch on resonant modes temporally for sensing if needed. ω_0 tuning leads to resonant modes that are always present, and

thus, for some wavelengths, the system is highly reflective. γ_{int} tuning introduces losses into the system and reduces optical transparency for resonant and off-resonant wavelengths. γ_{rad} -driven sensors can remain fully transparent in “off” mode, transmitting the entire spectral band, and then selectively switch “on” the resonance only when particular analytes need to be detected. This is especially advantageous in fiber-based optical systems, where the sensor element remains transparent during normal operation and can be optically activated to create a high-Q response for refractive index sensing.

3. Polaritonic Critical Coupling: Light-matter coupling strength for polariton formation depends on the balance between photonic and material losses. Typically, the photonic mode’s total loss $\gamma_{\text{photon}} = \gamma_{\text{rad}} + \gamma_{\text{int}}$ should match or be similar to the material excitation’s linewidth γ_{mat} to reach so-called polaritonic critical coupling.² Thus, there is a need to actively tune the photonic loss rates to probe and optimize this ratio. ω_0 tuning merely shifts the mode’s center frequency and cannot reduce or increase the total loss channel. γ_{int} tuning, meaning increasing intrinsic cavity losses, quenches near-field enhancement and can outweigh the material excitation, effectively preventing polariton formation. The γ_{rad} tuning advantage lies in keeping γ_{int} low and only altering the radiative part of the photonic loss so that it matches the material’s loss.

4. Emission Linewidth Control and Tunable Lasing: Many photonic devices rely on controlling the emission properties, such as linewidth and coherence, of an integrated gain medium. In many applications, there is a need to actively control these properties. ω_0 tuning shifts the resonant frequency of the photonic mode but does not fundamentally alter its damping, so the emission linewidth cannot be changed. γ_{int} tuning induces additional intrinsic losses and reduces the overall emission strength, as further nonradiative decay channels are introduced. γ_{rad} tuning preserves the low nonradiative intrinsic losses of the photonic mode while allowing the radiative decay rate and thus the linewidth to be freely adjusted. For lasing applications, the threshold condition depends on the total cavity losses, and if γ_{rad} is too low, it can be difficult to couple pump light into the cavity or out-couple coherent photons, while if γ_{rad} is too high, the threshold power increases. Dynamically controlling γ_{rad} allows optimization of the cavity’s Q-factor, enabling broad- and narrow-line lasing modes with adjustable thresholds.

5. Ultrafast Pulse Modulation and Time-Crystal Metasurfaces:

In the spectral vicinity of the quasi-BIC, significant dispersion is induced on propagating light. γ_{rad} based resonance tuning therefore leads to far-reaching control over key material properties on ultrafast timescales, which is a requisite for experimentally achieving time-crystal metasurfaces. Furthermore, it can be used for novel compact pulse-compression systems: By modifying the dispersion on the same timescale as the pulse, a temporally varying dispersion is induced, meaning the front of the pulse propagates at a different speed than its back. Therefore, this scheme controls

pulse lengths without splitting them into their spectral components, potentially allowing on-chip solutions.

Compared with γ_{int} tuning, this approach offers two main advantages: First, tuning γ_{rad} enables precise control over how long light stays in the cavity or when it leaks; γ_{int} -based tuning will only destroy the mode by increasing losses, which is unwanted in these applications. Furthermore, the measured metasurface allows for sub-picosecond increases and decreases in γ_{rad} depending on the exact geometry, hence leading to positive or negative tunable delays. For γ_{int} , on the other hand, ultrafast decreases remain elusive.

6. Nonvolatile Encoding of Metasurfaces: Phase change material-based encoding in metasurfaces can be used to store data or to adjust resonance parameters post-processing. Challenges with ω_0 or γ_{int} tuning arise because using phase-change materials like GST or Sb_2Se_3 for conventional tuning either shifts the resonance or introduces additional absorption, never fully toggling the mode from nonresonant (transparent) to resonant. γ_{rad} tuning, on the other hand, allows straightforward encoding. Starting with a metasurface in its dark state and using a localized optical pulse to partially change the phase in certain unit cells, resonances can be coupled to the farfield in a controlled manner. A subsequent erase step, for example, uniform heating, resets the material to its transparent state. Because γ_{rad} tuning can achieve a genuine “off” resonance, this leads to reconfigurable devices that stay transparent until selectively written, enabling optical storage, adjustable patterning of wavefronts, or post-processing optimization.

Reviewer comment:

The main point that concerns me is the fact that the authors focused on the modifications of the radiative losses while almost fully neglecting the non-radiative part. This sublimates into the following questions:

(i) The authors use TCMT to extract the radiative losses from the spectroscopy data. However, the reliability of this extraction relies heavily on the accuracy of the measured peak amplitude. In a realistic scenario, the measured transmission spectra would represent an average over a finite range of angles of incidence. Depending on the dispersion of the modes of the structure, this can lead to considerable broadening and suppression of the resonance. As I did not find a very detailed description of the measurement configuration, I could only assume that the spectra is measured with the supercontinuum probe covering the full NA of the objective (0.25). The authors should consider how this influences the accuracy of their estimations on γ_{rad} (and, by extension, γ_{int}).

Answer:

We thank the Reviewer for emphasizing the importance of discussing non-radiative losses. In our initial submission, we did not explicitly detail the sources of these losses. Indeed, far-field spectra alone cannot isolate the physical origin of individual non-radiative contributions forming γ_{int} . From an experimental standpoint, absorptive losses play a crucial role because, among the various non-radiative mechanisms, they are the only one affected by pump illumination. However, when discussing resonance modulation based on intrinsic loss, γ_{int} is the best parameter to describe such active tuning, as it is an observable in far-field spectra. In practice, it includes material absorption, scattering from inhomogeneities and surface roughness, fabrication-induced size variations, finite-array-induced losses, and the damping effects of any angular spread in illumination. While we will continue using γ_{int} in our analysis because it is a widely recognized parameter in the community, we will explicitly name these contributing factors in the text to avoid confusion.

Regarding the Reviewer's comment on the illumination angle, we want to note that the angular spread of the detected light is small, with only minor effects on mode broadening and damping compared to the other loss channels discussed above. Although our confocal setup uses a 0.25 NA objective for both the condenser and the objective, the actual angular spread is much smaller than the theoretical maximum collection angle of 14.5° . First, the objective pupil is 11 mm, but our beam diameter is only around 5 mm, reducing the maximum angle to 6.7° . Second, the actual spot size of the laser pulses on the sample and the area from which we collect the light further limit the angles involved.

In the newly added Supplementary Figure S24, panel (a) shows the pump illumination, which has a $49.5 \mu\text{m}$ beam diameter on the sample, achieved by highly defocusing the beam with an additional lens before the objective. Similarly, panel (b) shows the probe illumination, which yields an $18.8 \mu\text{m}$ spot. In both cases, because the beam is defocused, not all possible angles from -6.7° to $+6.7^\circ$ occur across each illuminated region. While angles can be higher at the sides, they approach zero near the center, where the illumination is nearly collimated. In contrast, the identical 0.25 NA objective used to collect the light is in focus, so the collection spot is estimated to be only about $2 \mu\text{m}$ in diameter. Overlaying this small collection region with the much larger defocused pump and probe spots makes it clear that primarily collimated (or almost collimated) light is actually collected.

As a result, the angular spread, which we can estimate to $\pm 0.7^\circ$, does not significantly affect the measured spectral width or modulation. As discussed below in the new supplementary note, simulations suggest a spectral shift of only up to 0.1 nm and 0.6 nm for TE and TM light, respectively. Such small shifts do not critically affect the modes studied in this work.

We thank the Reviewer for bringing up this important consideration, and we hope to have answered their concerns to a satisfying extent. Further, we invite the Reviewer to take a look at our response to Reviewer 2, response 6, where we discuss a newly added supplementary note showcasing the accuracy of our fits.

Action taken:

We added a sentence on page 7 regarding the different contributions to γ_{int} .

Note, the initial value of γ_{int} is composed of multiple contributions, including material, surface roughness, and finite array size losses.

We added Figure S24 showcasing the illumination schemes as well as a sentence linking to it in the main manuscript on page 14.

This configuration leads to only a minimal angular spread of the collected light, keeping the angular spread-induced broadening to a minimum, see **Supplementary Note 13**.

Furthermore, we added a new supplementary note discussing the influence of angled illumination and collection and its effect on the measured spectra.

Supplementary Note 13: Influence of Angled Excitation and Collection

To evaluate the effect of angled illumination and collection on the measured data, we analyze the configuration of our confocal setup in detail. Both excitation and collection use 0.25 NA objectives, whose theoretical maximum half-angle is 14.5° . However, for the reasons discussed below, the practical angular spread is significantly smaller. First, although the objective pupil diameter is ~ 11 mm, our beam diameter is only ~ 5 mm, reducing the maximum incidence angle to 6.7° . Second, the pump and probe beams are deliberately defocused, resulting in large spot sizes on the sample: $49.5 \mu\text{m}$ for the pump (**Figure S24a**) and $18.8 \mu\text{m}$ for the probe (**Figure S24b**). Since not all rays within these spots contain the full $\pm 6.7^\circ$ angular spread, the center of each spot is nearly collimated, while higher angles appear only at the edges. In contrast, the 0.25 NA objective used for collection is in focus, so the sample region from which we collect light is only $\sim 2 \mu\text{m}$ in diameter (**Figure 24c**). Overlaying this small collection region with the much larger defocused spots reveals that mostly collimated, or nearly collimated, rays are collected. We can estimate the angular spread in this central collection region using geometric optics and a small-angle approximation. As the angular range across the full $18.8 \mu\text{m}$ probe spot goes from $\pm 6.7^\circ$, the local angle at a radius r is approximately $\theta(r) = \theta_{\text{exc,max}}/r_{\text{exc}}r$ where $r_{\text{exc}} \approx 9.4 \mu\text{m}$ is the spot radius and $\theta_{\text{exc,max}} \approx 6.7^\circ$. Evaluating at $r = 1 \mu\text{m}$ (the radius of the collection region), the

maximal local angle becomes $\theta(1\mu\text{m}) \approx 0.71^\circ$. Simulations show that for both TE- and TM-polarized light, this leads to minor angular shifts of <0.1 and 0.6 nm, respectively. As the studied Q-factors are all substantially below 500 (FWHM around 1.5 nm), these angular spreads have only a minor effect on mode broadening compared to other experimental loss factors like scattering on rough surfaces or varying resonator sizes.

Figure S24: Illumination and Collection Scheme. Both objectives in the confocal setup have a 0.25 NA with an 11 mm pupil. Since the laser beam diameters are only around ~ 5 mm, the effective pupil diameter is smaller, limiting the maximum incidence angle to 6.7° . (a) The pump excitation scheme shows a highly defocused beam, giving a $\sim 49.5 \mu\text{m}$ full-width at half-maximum (FWHM) spot on the sample. (b) The probe beam is similarly defocused, yielding an $\sim 18.8 \mu\text{m}$ spot. Both beams pass through the same objective, but their degrees of defocus are independently adjusted with additional lenses. (c) The collection objective is in focus, with an estimated $\sim 2 \mu\text{m}$ collection area. Although the objective can collect a broader range of angles, a confocal aperture in front of the spectrometer further restricts the light cone.

Reviewer comment:

(ii) The authors claim that the optical pumping they provide changes the radiative losses, while “the effect on γ_{int} is negligible” (page 3, line 89). This needs a more detailed discussion. In fact, the photogenerated free carriers contribute significantly to both the real and imaginary parts of the refractive index. In the short wavelength range, the changes to n and k could be of comparable magnitude. The observed spectral shifts of the resonance (up to 10 nm) indicate quite strong change of n , which implies considerable change in k . Figure S11 reveals that the extracted non-radiative losses are almost 20x larger than the radiative part - what authors probably imply is that the changes to γ_{int} are small with respect to its original value; this needs to be detailed earlier. It is particularly important in the context of the outlook that the authors provide in the conclusion (...This is in contrast to conventional ω_0 and γ_{int} tuning methods, which struggle to

dynamically control local field enhancement or spectral width without introducing significant parasitic losses...). Both the local field enhancement and linewidth are defined by the balance of radiative and non-radiative losses, and with the initial strong imbalance of the two (at least, based on the measured data) I don't see an obvious benefit of controlling the radiative part.

Answer:

We agree with the Reviewer that the discussion on γ_{int} needs to be more in depth and earlier in the manuscript. We moved the discussion on γ_{int} from the old Figure 5 to Figure 3, right at the point where we show pump-probe results for the very first time, see new Figure 3f. We also clarify that by “only minor changes to γ_{int} ,” we mean that while γ_{int} does indeed change under pumping, its relative change is small compared to its original value (as the Reviewer has noted as well).

Regarding a discussion on the physical origin of the fast increase of γ_{int} (including photogenerated free carriers), we refer the Reviewer to our answer to their question (iii)

Regarding the expected change of the real and imaginary part of the refractive index, we refer the Reviewer to our response to their question (iv), which is now also detailed extensively in the new Supplementary Note 11. Notably, the expected change of the real part of the refractive index is much higher than the change of the imaginary part.

Regarding the Reviewer's question on the balance of γ_{int} and γ_{rad} for local field enhancement (FE): Our experiments are all conducted well within the undercoupled regime. There, γ_{int} significantly exceeds γ_{rad} . Consequently, increasing γ_{rad} so the system is tuned towards the critical coupling condition directly raises the FE, granting effective control over it. Changes to γ_{int} in this operational region do not change this direct relation of γ_{rad} with the FE. This would only change close to the critical coupling region where the FE is maximal.

Following Maier et al.,³ we estimate the local field enhancement using $FE \propto \sqrt{\gamma_{\text{rad}}}/(\gamma_{\text{rad}} + \gamma_{\text{int}})$. To illustrate this, we analyzed the case in Figure 3e-f. We normalize the initial FE to 1 as a baseline. Upon pumping, and using γ_{rad} and γ_{int} from Figure 3f, we observe in Figure S19 that:

- Within the first 300 fs, there is a drop in FE due to high intrinsic losses coinciding with the pump-probe temporal overlap.
- After that short window, the FE rises by approximately 50% and then slowly decays back to its initial value over the subsequent 40 ps.

This demonstrates the direct impact of the pump on the FE. Naturally, if $\gamma_{\text{rad}} = 0$ initially, the FE would start at a small value and jump to a large value, representing an even more dramatic relative

change. We thank the Reviewer for highlighting this relationship between FE and both loss rates, and we believe our new data on FE evolution under pumping is a valuable addition to the manuscript, now shown in Supplementary Note 10.

Action taken:

We moved the discussion of γ_{int} from Figure 5 to Figure 3, the very first time we show time resolved results. We further added a specific statement that only the relative shift of δn is large compared to the one of γ_{int} .

...

While non-resonant pumping is shown in **Supplementary Note 7**, **Figure 3e** shows a time trace for a 720 nm y-polarized pump pulse on resonance with Mie 1. A spectral shift of 9.2 nm is observed, which decays to the original spectral position with a time constant of around 20 ps. Furthermore, after the pump, the resonance amplitude shows a clear increase, indicating a significant change in γ_{rad} due to the experiments being performed in the undercoupled regime. Using the established TCMT model (**Supplementary Note 1**), we fit the transient spectra (**Supplementary Note 9**) and extract the time-dependent loss rates shown in **Figure 3f**. Note, the initial value of γ_{int} is composed of multiple contributions, including material, surface roughness, and finite array size losses. Around 1 ps after the pump, both γ_{int} and γ_{rad} are increased by roughly 14% and 250%, respectively, and decay within 20 ps. In addition to the carrier concentration, γ_{int} is also susceptible to two-photon absorption, electron temperature and mode shifting, leading to deviating decay behavior from γ_{rad} (see **Supplementary Note 10**). Because the sustained modulation is dominated by γ_{rad} , the following analysis focuses on modeling the radiative loss.

...

Figure 3: Mie modes and selective resonant pumping. (a) Experimental transmittance spectrum at the RSP-BIC condition with $l_2 \approx 226$ nm for x- and y-polarized light, revealing two and one Mie modes, respectively (labeled Mie 1, Mie 2, and Mie 3). (b) The unit cells' normalized power loss density map (energy loss per unit volume in W/m^3) for the three Mie modes shows distinct mode profiles and dissimilar losses in both rods at a cutting plane of $z = 30$ nm. (c) Comparison of average power loss density between the two rods for each Mie mode. Mie 1 exhibits the highest ratio between Rod 1 and Rod 2, indicating the strongest selective absorption. (d) Sketch of the optical pumping principle: Above-bandgap excitation of carriers from the conduction band (CB) to the valence band (VB) generates free electrons and holes, altering the polarizability and the refractive index. (e) Pump-probe spectral time trace (720 nm pump with a fluence of $100 \mu\text{J}/\text{cm}^2$) for a metasurface with $l_2 = 236$ nm. The pump pulse is y-polarized and the probe pulse is x-polarized, leading to 9.2 nm spectral shift of the SP-BIC and an increase in resonance amplitude. (f) Corresponding γ_{int} and γ_{rad} obtained using TCMF fitting. A sharp increase of γ_{int} by approximately 100% upon pump arrival, followed by a rapid drop to approximately 14% above pre-pump values within 1 ps is visible. γ_{rad} increases by 250%, remaining at higher values before it returns back to prepump values within 20 ps. The inset on the right reveals the quick decay of intrinsic losses within the first ps after pump arrival.

...

Furthermore, we added a section in Supplementary Note 10 alongside with Figure S19 and a linking text in the main manuscript on the influence of γ_{rad} on the field enhancement. The Supplementary Note 10 is reprinted in the following response to question (iii), and the linking text is reprinted in our first action for this question.

Reviewer comment:

(iii) The apparent uncertainty in the extracted γ_{rad} and γ_{int} is particularly important in the context of the discussion of data in Figs. 4 and 5. The authors attribute the change of refractive index solely to the contribution of photogenerated free carriers. If that is the case, the relaxation time of the real and imaginary part of the refractive index should be similar. Instead, what the authors observe is a fast decay of γ_{int} that they attribute (quite vaguely) to nondegenerate photon absorption and to the cavity resonance lifetime. However, why does the longer cavity lifetime contribute to the non-radiative but not radiative losses?

Answer:

We thank the Reviewer for this in-depth observation - changes in intrinsic and radiative losses are indeed not directly correlated. There are multiple reasons for this:

First, the intrinsic/non-radiative loss does not just describe how the material absorbs radiation; in fact, it combines all loss channels that do not arrive in the observation port, so also phenomena like scattered radiation. Generally, these loss channels lead to a similar effect on the resonance's lineshape, and can therefore be distinguished from radiative losses through a fit model. The broadening caused by the rapid resonance shift (and its associated reduction in amplitude) is mostly in line with the effect of non-radiative losses, and therefore does not change the radiative loss in our model.

Second, as the probe wavelength is still in the range of low absorption in silicon, its absorption can be increased in the presence of the pump beam due to nondegenerate two-photon absorption (TPA), meaning a combination of one pump photon and one probe photon are absorbed. This can only happen in the same timeframe as the spectral shift, and not during the following slow decay. In an unstructured film, with a peak intensity of $5 \times 10^8 \text{ W/cm}^2$ and a TPA coefficient around 850 nm of 2 cm/GW ,⁴ one would expect a small change of $\Delta k = 6 \times 10^{-6}$. Due to the unknown field enhancement inside the structure of both pump and probe beam, which amplifies this effect, the exact magnitude is hard to estimate in our case.

Third, electron-electron scattering processes can happen on sub-picosecond timescales and influence probe absorption.

As these effects show, photogenerated free carriers are indeed not the only contribution to changes in intrinsic losses; both are independent from increased losses due to excited carriers. However, because our newly added theory predicts refractive index changes around 0.18, alternative mechanisms for changes in the refractive index (and therefore radiative losses) can be neglected at our pump intensities. Specifically, the Kerr effect or lattice distortion effects like electrostriction would be significantly smaller, and the low repetition rate excludes thermal effects (the observed resonance behavior in our experiments was indeed invariant with changes in the repetition rate). This explains the different decay dynamics.

Furthermore, we agree on the importance of the uncertainty of the extracted loss rates. Mean extracted errors of our fits are below 4% in all cases; for an in-depth evaluation of the CMT fits, see our answer to Reviewer 2, comment 6, including 3 new figures and one Supplementary Note.

Action taken:

To clarify our errors in the CMT fits, we added Supplementary Note 9 (see Reviewer 2, comment 6). The estimated mean error for γ_{rad} is around 2.4%, while the error for γ_{int} is 3.6%, demonstrating the high fitting accuracy.

Further, in Supplementary Note 10, we explain the temporal evolution in detail including a discussion on the faster decay of γ_{int} .

Linking text on page 7 and 9

In addition to the carrier concentration, γ_{int} is also susceptible to two-photon absorption, electron temperature and mode shifting, leading to deviating decay behavior from γ_{rad} (see **Supplementary Note 10**).

Supplementary Note 10 shows a corresponding power series, demonstrating continuous γ_{rad} tunability.

This causes a *sharpening* of the resonance, with a decrease in amplitude, while γ_{rad} drops from 0.35 to 0.12 THz, and the total Q-factor increases by 150% from around 100 to 250 (**Supplementary Note 10**).

Supplementary Note 10: Temporal resonance evolution

Pump-induced changes in the radiative loss rates of the metasurfaces affect both width and depth of the resonance, and are accompanied by changes in non-radiative losses. To visualize the overall performance of the pump to change the mode, we therefore first take a closer look at the combined Q-factor of the resonance in the broadening and sharpening condition (**Figure S17**). In the former case, the Q-factor decreases from an initial value of 400 down to 200 upon temporal overlap of the pump and probe pulses. It then stabilizes around 300 before gradually increasing back to its initial value of 400. In the latter case, the Q-factor rises from an initial 110 to 250, then gradually converges back to the initial value.

Figure S17: Total Q-factors During Mode Broadening and Sharpening. (a) Total Q-factor obtained using the TCMT model with variable γ_{rad} and γ_{int} for the transmittance time trace shown in **Figure 5b** with $l_2 = 236$ nm. (b) Equivalent representation to (a), but for $l_2 = 186$ nm and the time trace of **Figure 5e**.

As discussed in **Supplementary Note 5**, changes in the refractive index in rod 1 give rise to a change in radiative properties of the mode. A close examination of the observed behavior allows to draw conclusions on the physical processes involved in the modulation and observed decay, but requires differentiating between radiative and non-radiative losses.

First, analogous to **Figure 3f**, we now take a look at the absolute values of the two loss rates for an exemplary geometry that enables an increase in modulation depth upon pumping. As seen in **Figure S18**, γ_{rad} first increases in a pulse-length limited timeframe, and decays with a time constant <10 ps, with some deviations from pure exponential decay as expected due to the nonlinear relationship between Δn and γ_{rad} in the broadening case (see **Supplementary Note 5**).

Since the modulation remains unchanged for lower repetition rates, thermal effects can be excluded. Furthermore, we see no pulse-length limited peak, rendering the Kerr-effect insignificant. This is expected from our structure needing a refractive index change of 0.18 to achieve the shown shift (see **Supplementary Note 11**), which is beyond what is achievable at our intensities with Kerr of lattice distortion effects like electrostriction, but can be realized with laser-induced carrier concentrations in the order of 10^{20} cm^{-3} ,⁵ which is below what is observed in laser ablation experiments in silicon.⁶ Lattice distortion effects might however contribute to minor longer-lasting changes in the radiative losses.

γ_{int} , on the other hand, is significantly larger than γ_{rad} at the start of the experiment, which shows that the structure operates in the undercoupled regime. Consequently, changes in absorption due to the increased carrier concentration in the conduction band after pumping only gives rise to a minor relative change in γ_{int} . However, we observe a large spike in γ_{int} , that is limited by the pulse duration. This is attributed to three effects: First, as the probe pulse around 800 nm is still in the low-absorption range of crystalline silicon, nondegenerate two-photon absorption can temporarily increase nonradiative losses while the pump pulse is also present in the sample. Second, the rapid shift of the resonance on timeframes shorter than our probe pulse causes an effective mode broadening through both the probe pulse being too long to resolve this process, as well as now non-resonant optical radiation scattering out of the resonator. Third, electron-electron scattering processes happen on timescales of similar length as our pump pulse, and can lead to ultrafast changes in absorptive properties.

Remarkably, γ_{int} first decreases to a value around 14% above its baseline after this initial spike, and continues increasing for the first 10 ps after the pump arrives. We attribute this slow rise to the cooling of the hot electron distribution, which happens in the timescale of a few picoseconds⁶ and can impact absorptive properties. Only the subsequent decay is then analogous to the carrier decay in the refractive index modulation.

Figure S18: Radiative and Intrinsic Loss Comparison. For $l_2 = 236$ nm, the transmittance time trace is fitted using the TCMT model with variable γ_{rad} and γ_{int} . (a) displays γ_{rad} , while (b) shows γ_{int} .

This extraction further allows gaining insights in the evolution of the field enhancement (FE) within the structure by estimating $FE \propto \sqrt{\gamma_{\text{rad}} / (\gamma_{\text{rad}} + \gamma_{\text{int}})}$ (see **Figure S19** for the FE from **Figure S18**). As this metric is a combination of radiative and non-radiative losses, we see effects from both loss rates: The initial spike in γ_{int} causes an ultrafast drop in the field enhancement, while the later increase and decay are mostly governed by the evolution of γ_{rad} due to the small relative change in γ_{int} .

Figure S19: Estimated FE change. For $l_2 = 236$ nm, the previously extracted γ_{rad} and γ_{int} are combined in an effective FE.

To gain further insight in decay dynamics and the absorption mechanism, we now investigate pump power dependencies. As seen in **Figure S20**, both radiative losses and the mode change at 1 ps after the pump arrives: in the case of resonance broadening, γ_{rad} increases, and the mode increases in amplitude and width. The spectral position scales approximately linearly with the pump power, showing mostly linear absorption causing the refractive index modulation, while the continuity in γ_{rad} furthermore demonstrates continuous tunability.

Figure S20: Power dependence at $t = 1$ ps. (a) Pump-fluence dependent transmittance spectrum for $l_2 = 226$ nm (case of resonance creation), showing a gradual increase in resonance amplitude with increasing pump fluence. (b) Corresponding γ_{rad} values derived from fits to the data in (a), demonstrating a continuous increase in γ_{rad} with fluence, reaching up to 0.11 THz.

Decay times of the spectral shift and the radiative loss modulation are shown in **Figure S21a,b**. The latter is generally shorter than the decay time of the spectral shift because the radiative loss depends mostly quadratically on the asymmetry. Furthermore, because of Auger recombination at high carrier densities, the excited populations in the two rods decay at different rates, causing their asymmetry to decay faster than the overall refractive index modulation. However, due to the minor decrease in decay times of the spectral shift at the highest intensities ($< 20\%$), Auger processes only play a minor role in the observed dynamics.

Figure S21: Resonance Decay Times. (a) $1/e$ decay time of the pump-induced spectral shift in dependence of the incident pump fluence for the broadening case. Decreasing decay times at high fluences suggest an increasing influence of Auger recombination. (b) $1/e$ decay time of the fitted radiative loss of the resonance.

Combining these observations allows us to model the decay of the refractive index modulation in good approximation with a single exponential reflecting mostly surface recombination, at a time constant $\tau = 20$ ps for low excitation fluences.

Reviewer comment:

(iv) To my opinion, the paper will benefit a lot from the estimations of the induced refractive index changes and a connected discussion of the limitations of the approach for creation/annihilation of more pronounced optical resonances.

Answer:

We thank the Reviewer for highlighting the need to clarify the expected refractive index changes in our measurements. To address this, we have added the new Supplementary Note 11 that estimates the refractive index change by numerically reproducing the experimentally observed resonance wavelength shift and Q-factors. Following our selective pumping approach, we change the refractive index in both resonators, but with a 3.5:1 ratio (based on Figure 3c), since the resonant rod (rod 1) experiences a stronger shift than the non-resonant rod (rod 2). As shown in the new note, we estimate a change of $\Delta n_{\text{rod1}} = -0.18$ in rod 1 and $\Delta n_{\text{rod2}} = -0.05$ in rod 2, leading to a refractive index contrast between the two resonators of $\Delta n = -0.13$.

While estimating Δn is fairly straightforward, determining Δk is more challenging. In simulations, only intrinsic losses can be directly incorporated; finite-array effects, inhomogeneities, and surface roughness are not easily modeled. Nonetheless, we varied k under two scenarios: $\Delta n = 0$ and $\Delta n = 0.13$ in rod 1 (again maintaining a 3.5:1 ratio of a change in k between rods). We then compared the fitted Q-factors from these simulations to the experimental results to determine Δk . Strikingly, we found the change of k is 0.004, noticeably smaller than Δn .

We further acknowledge the limitations of silicon-based tuning caused by initial intrinsic losses as well as pump induced losses, which we now discuss in both the results and conclusion sections. However, our method is not restricted to silicon or this particular tuning mechanism. As noted in our conclusion, it can be readily applied to systems employing phase-change materials or Kerr-type materials with potentially much lower losses.

In order to create more pronounced optical resonances using our active tuning approach, multiple optimizations can be made, all revolving around reducing initial intrinsic losses. By using higher resolution lithography (e.g. increasing the acceleration voltage from 30 kV to 200 kV, using thinner layers of resist, and decreasing the line distance of the written structure), the final metasurface may be more homogeneous and the structures smoother, creating less scattering loss. Further, by increasing the array size, we expect that losses caused by the finite array size will be reduced. Next, the current structure is placed on top of a sapphire substrate and embedded in a SiO₂ layer made by spin on glass. We believe that the thick, but in the single digit μm regime thick SiO₂ height may cause some additional interference based losses. To further reduce losses, a homogeneous embedment within sapphire only with a very high layer height would increase

efficiency. Lastly, after all these optimizations are performed, we reach a regime where also the illumination plays a substantial role regarding the remaining losses (previously we deemed them non-dominating in our current configuration). Thus, one should consider perfectly collimated pump and probe illumination. Lastly, one can shift the resonance to larger wavelengths where initial values are lower, further increasing the signal modulation.

In response to the Reviewer's call for a deeper understanding on the pumping-induced changes of γ_{int} and γ_{rad} , we developed an extensive resonant-state-expansion (RSE) model. The new model (now presented in a completely new Figure 4) shows:

1. The RSE reproduces the experimentally observed wavelength shift and reveals that an index perturbation mixes the antiparallel-dipole qBIC with the parallel-dipole (PD) resonance.
2. A function for the RSP-BIC Q-factor-perturbation relation, deviating from the usual inverse-square law.
3. RSE-predicted values of γ_{rad} for broadening, annihilation, and sharpening match the measurements, yielding an index change of $\Delta n \approx 0.1$ in the pumped rod.

Figure 4 replaces the previous Figure 4 (now Figure 5) and provides an intuitive map: depending on the starting length l_2 , the same Δn drives γ_{rad} either up or down, enabling the four switching modes confirmed experimentally in Figure 5.

Action taken:

We added a Supplementary Note 11 including two figures to discuss the change of n and k while pumping.

Supplementary Note 11: Estimated refractive index changes

To estimate how both the real (n) and imaginary (k) parts of the refractive index change under laser pumping, we reproduced the mode shift and broadening observed in **Figures 4b** and **5a,b** through numerical simulations. Based on **Figure 3c**, the absorbance, and thus the changes in n and k , is approximately 3.5 times higher in Rod 1 than in Rod 2 once pumped, as illustrated in **Figure S22a**.

Step 1: Estimating Δn : We first determine the change in the real part of the refractive index from the shift of the resonance wavelength. In the simulations, silicon is assigned a baseline $n = 3.67$ and $k = 0.006$ (following Schinke et al.⁷ around 790-800 nm). To precisely match the unpumped experimental resonance wavelength (i.e., $t = -1$ ps) with the simulated resonance wavelength,

we slightly reduce the in-plane dimensions of the unit cell by 2.3%. **Figure S22b** shows the comparison between the experimental (dashed gray curve) and simulated (solid gray curve) spectra. Next, we tune Δn_{rod1} (and hence $\Delta n_{\text{rod1}} \approx 3.5 \Delta n_{\text{rod2}}$) until the pumped resonance wavelength at $t = 1$ ps spectrally matches with the experimental results (blue curves). This causes a total shift of about 10 nm, which we reproduce numerically using $\Delta n_{\text{rod1}} = -0.18$. Sweeping Δn_{rod1} from 0 to -0.18 continuously shifts the resonance and simultaneously increases its amplitude, consistent with rising radiative loss γ_{rad} as shown in **Figure S22c**.

Step 2: Estimating Δk : With $\Delta n_{\text{rod1}} = -0.18$ ($\Delta n_{\text{rod2}} = -0.05$, hence $\Delta n = |\Delta n_{\text{rod1}} - \Delta n_{\text{rod2}}| = 0.13$) fixed, we estimate the accompanying change in the imaginary part of the refractive index. Because the simulation only accounts for intrinsic material loss, we first determine an effective $k = k_{\text{est}}$ that reproduces the pre-pump non-radiative loss γ_{int} measured experimentally (0.94 at $t = -1$ ps, **Figure S18**). Sweeping the same k in both rods (**Figure S23a,b,c**) shows γ_{int} varying linearly with k while γ_{rad} is unaffected; a linear fit yields $k_{\text{est}} = 0.022$ (black line), up from the literature Si value $k = 0.006$ (Schinke et al. ⁷). To find the pump-induced change we assume linear absorption and the same 3.5:1 imbalance as for Δn , i.e. $\Delta k_{\text{rod1}} = 3.5 \Delta k_{\text{rod2}}$ and $\Delta k = \Delta k_{\text{rod1}} - \Delta k_{\text{rod2}}$ (**Figure S23a**). Using the post-pump value $\gamma_{\text{int}} = 1.04$ at $t = 1$ ps (**Figure 3e**), the linear fits in **Figure S23d,e** give $\Delta k_{\text{rod1}} \approx 0.004$ (blue line, $\Delta k \approx 0.003$). Thus, the pump increases k by only a few 10^{-3} , consistent with our observation that the index contrast is dominated by Δn , while Δk contributes only a minor fraction of the total modulation.

Figure S22: Estimating Δn . (a) From **Figure 3c**, pumping leads to a 3.5 times higher absorbance in Rod 1 than in Rod 2, leading to $\Delta n_{\text{rod1}} \approx 3.5 \Delta n_{\text{rod2}}$. (b) Experimental spectra at $t = -1$ ps and $t = 1$ ps (dashed gray and blue, respectively) compared to the simulations (solid gray and blue). The in-plane dimensions of the simulated structure are reduced by 2.3% to align spectrally with the unpumped experimental mode. Next, Δn_{rod1} is swept from 0 to -0.18 to match with the pumped resonance wavelength. (c) Full Δn_{rod1} sweep from 0 to -0.18 , showing a continuous 10 nm spectral shift and an amplitude increase matching the experiment.

Figure S23: Estimating Δk from pump-probe data. (a) Schematic: besides an initial k value in both rods the pump generates additional absorption with a 3.5:1 imbalance, so $\Delta k_{\text{rod1}} \approx 3.5 \Delta k_{\text{rod2}}$. (b,c) Pre-pump calibration. Sweeping the intrinsic index k from the literature value 0.006 (grey line) to 0.10 leaves γ_{rad} unchanged (b) but increases γ_{int} linearly (c), matching the experimental $\gamma_{\text{int}} = 0.94$ at $t = -1$ ps (**Figure 3e**) gives $k_{\text{est}} = 0.022$ (black line). (d,e) Post-pump estimate. With k fixed at 0.022, Δk_{rod1} is swept from 0 to 0.025. Again γ_{rad} is unaffected (d), while γ_{int} rises linearly (e); the experimental $\gamma_{\text{int}} = 1.04$ at $t = 1$ ps (**Figure 3e**) is met for $\Delta k_{\text{rod1}} \approx 0.004$ (blue line), corresponding to $\Delta k \approx 0.003$.

We added a linking text on page 7.

To quantify the pump-induced refractive index change, we match the experimentally observed resonance shift of 9.2 nm to numerical simulations (**Supplementary Note 11**), yielding a refractive-index difference between the two rods of $\Delta n = n_{\text{rod2}} - n_{\text{rod1}} = 0.13$ (**Figure 4a**).

We added a short section discussing the limiting factor on the modulation depth in Supplementary Note 3

Future improvements to increase Q-factors and modulation strength would consist of reducing surface roughness by increasing the acceleration voltage of the e-beam, increase the array size to reduce edge leakage, replace the SiO₂ layer by an even thicker sapphire layer, or shift the resonant mode towards the near infrared to reduce material absorption.

Furthermore, we added Figure 4 and a corresponding section to the main text, as well as Supplementary Note 5 on the RSE model.

Supplementary Note 5: Resonant State Expansion

To gain more insights into the mechanism of changing eigenmodes during optical pumping, we implement the Resonant State Expansion (RSE) theory and analyze how a change in the permittivity by $\Delta\epsilon$ of the rods affects the RSP-BIC. For simplicity, we assume a constant refractive index of crystalline silicon, $n_0 = 3.7$ and place our metasurface in vacuum to eliminate any parasitic diffraction channels associated with the substrate.

First, we analyze the eigenmodes of the metasurface in this modified environment using the Electromagnetic Waves Frequency Domain module in COMSOL Multiphysics. **Figure S6a** shows a colormap of the qBIC Q-factor as a function of the width of the first rod w_1 and the length of the second rod l_2 . All other geometrical parameters remain the same as in the main text: square unit cell with period 420 nm, $l_1 = 175$ nm, $w_2 = 95$ nm, height of the rods 115 nm. In this 2D w_1 - l_2 parameter space, the RSP-BIC condition is represented by a continuous line. Due to highly confined, high Q-factor qBIC mode, the parameters for RSP-BIC we chose ($w_1 = 185$ nm and $l_2 = 210.5$ nm), closely match those of the original structure on a sapphire substrate embedded in SiO₂ ($w_1 = 185$ and $l_2 = 216$ nm). The new RSP-BIC has a resonance wavelength of 637 nm and the complex eigenfrequency ω_2 . **Figure S6b** shows that increasing either parameter shifts the resonance to longer wavelengths.

RSE is a powerful tool to describe the hybridization of a finite number of metasurface eigenstates by introducing a perturbation. By changing $\Delta\epsilon$ in the first rod, we preserve all three mirror symmetry planes along the main axes, so an antiparallel dipole quasi-BIC, whose electric field is predominantly along the x direction and has the components $E_{x,y}(x, y, z) = E_{x,y}(x, y, -z)$ (plane $z = 0$ is across the middle-cut of the nanorods), can only couple with an eigenstate of the same parity. The closest candidate is a parallel dipole (PD) resonance at 495 nm with the complex eigenfrequency ω_1 . In **Figure S6c** we plot the metasurface's x- and y-polarized transmittance spectra and show corresponding eigenmodes at the marked spectral positions. For clarity, we classify eigenmodes by current distribution over a mid-plane cross-section of the nanorods in the xy-plane as either "electric" (see **Figure S6d**) or "magnetic" (see **Figure S6e**). Due to removing the substrate and coverage, the quasi-BIC has no transparent background and lies near the "magnetic" resonance I. However, such a change does not affect our subsequent eigenstates analysis.

We define super-vectors \mathbf{F}_1 and \mathbf{F}_2 , $\mathbf{F}_i(\mathbf{r}) = \{\mathbf{E}_n(\mathbf{r}), i\mathbf{H}_n(\mathbf{r})\}$ for the PD resonance and antiparallel dipole quasi-BIC, respectively. To apply RSE to metasurfaces as open optical resonators, the states with eigenfrequencies ω_n must be correctly normalized. We perform the normalization described in:⁸

$$1 = \|\mathbf{F}_n\|^2 = \int_V [\epsilon(\mathbf{r})\mathbf{E}_n \cdot \mathbf{E}_n - \mathbf{H}_n \cdot \mathbf{H}_n] dV + \frac{ic}{\omega_n} \int_{\partial V} [\mathbf{E}_n \times (\mathbf{r} \cdot \nabla)\mathbf{H}_n + \mathbf{H}_n \times (\mathbf{r} \cdot \nabla)\mathbf{E}_n] \cdot d\mathbf{S}, \quad (1)$$

where V is the volume of a unit cell with surface boundaries ∂V . We also consider nonmagnetic materials with the permeability $\mu = 1$.

We introduce a variation $\Delta\varepsilon$ in the first rod as an environmental perturbation and use first-order perturbation theory based on the RSE. The hybridized states $\tilde{\mathbf{F}}_n$ are expressed as a linear superposition of the states \mathbf{F}_1 and \mathbf{F}_2 of the unperturbed system with $\Delta\varepsilon = 0$:

$$\tilde{\mathbf{F}} = \sum_{m=1}^2 a_m \mathbf{F}_m, \quad (2)$$

where we have truncated the RSE matrix equation to the two relevant states and keep coefficients a_n as a solution of a system of equations valid in the first order:

Figure S6: Analysis of eigenstates of the metasurface without perturbation. (a) Q-factor and (b) resonance position of the quasi-BIC as functions of the geometric parameters w_1 and l_2 . The logarithmic colormap of the Q-factor shows the condition for the RSP-BIC as a continuous line in the w_1 - l_2 parameter space. (c) Transmittance spectra of the metasurface for x and y-linearly polarized light. Labels indicate the positions of the “electric” and “magnetic” eigenstates, whose electric and magnetic field distributions over the center cut of the nanorods are shown in (d) and (e), respectively.

$$\omega_n a_n = \tilde{\omega} \sum_{m=1}^2 (\delta_{nm} + V_{nm}) a_m, \quad (3)$$

In the Eq.(3) δ_{nm} is the Kronecker delta and elements of the perturbation matrix read as:

$$V_{nm} = \int_V \Delta\varepsilon(\mathbf{r}) \mathbf{E}_n(\mathbf{r}) \cdot \mathbf{E}_m(\mathbf{r}) dV, \quad (4)$$

where the integration is performed by a volume of the first rod V with environmental perturbation $\Delta\varepsilon$.

Assuming constant $\Delta\varepsilon$ we introduce new parameters to the perturbation matrix:

$$v_1 = \int_V \mathbf{E}_1(\mathbf{r}) \cdot \mathbf{E}_1(\mathbf{r}) dV, \quad v_2 = \int_V \mathbf{E}_2(\mathbf{r}) \cdot \mathbf{E}_2(\mathbf{r}) dV, \quad u = \int_V \mathbf{E}_1(\mathbf{r}) \cdot \mathbf{E}_2(\mathbf{r}) dV. \quad (5)$$

By solving Eq.(3), we find a doublet of hybrid eigenfrequencies:

$$\omega_{\pm} = \frac{1}{2(1 - \tilde{u}^2)} \left[\tilde{\omega}_1 + \tilde{\omega}_2 \pm \sqrt{(\tilde{\omega}_1 - \tilde{\omega}_2)^2 + 4\tilde{u}^2 \tilde{\omega}_1 \tilde{\omega}_2} \right], \quad (6)$$

where the frequency parameters $\tilde{\omega}_{1,2} = \omega_{1,2} [1 + v_{1,2} \Delta\varepsilon]^{-1}$ describe the spectral shift produced by the perturbation and the parameter $\tilde{u} = u \Delta\varepsilon / \sqrt{(1 + v_1 \Delta\varepsilon)(1 + v_2 \Delta\varepsilon)}$ determines the mixing of the original eigenstates.

We empirically find that the absolute values of the parameters of the perturbation matrix (5) of normalized states \mathbf{F}_1 and \mathbf{F}_2 do not exceed 0.03. Moreover, all the parameters (5) are always multiplied by an environmental permittivity perturbation $\Delta\varepsilon$ in accordance with (4). Therefore, we expand the expression from Eq.(6) as well as \tilde{u} and $\tilde{\omega}_{1,2}$, neglect all cubic terms in $\Delta\varepsilon$ and obtain the final expression for the hybrid eigenfrequencies:

$$\omega_{PD} \approx \omega_1 \left[1 - v_1 \Delta\varepsilon - \frac{u^2 \omega_1 (\Delta\varepsilon)^2}{\omega_1 - \omega_2} \right] \quad (7)$$

$$\omega_{qBIC} \approx \omega_2 \left[1 - v_2 \Delta\varepsilon - \frac{u^2 \omega_2 (\Delta\varepsilon)^2}{\omega_1 - \omega_2} \right] \quad (8)$$

Having the analytical expressions Eqs.(7) and (8), we can now evaluate how introducing $\Delta\varepsilon$ in the first rod hybridizes the eigenmodes. According to our findings from **Figure 3f**, dissipation losses γ_{int} , compared to the radiative losses γ_{rad} , return to their original values faster after optical pumping. Therefore, we consider only the real part of $\Delta\varepsilon$, given by $\Delta\varepsilon = (n_0 - \Delta n)^2 - n_0^2 \approx -2n_0 \Delta n$.

Figure S7: Analysis of hybrid eigenstates of the metasurface with environmental perturbation Δn . Resonance position and the imaginary part of the eigenfrequencies (γ_{rad}) as functions of the perturbation Δn of the (a) RSP-BIC and (b) PD modes, obtained via COMSOL Multiphysics (solid lines) and RSE theory according to Eqs. (7) and (8) (dashed lines). (c) Q-factor of the RSP-BIC as a function of Δn obtained from both approaches. (d) γ_{rad} as a function of the second rod length l_2 for both the stationary case ($\Delta n = 0$) and perturbed case ($\Delta n = 0.13$). Arrow marks indicate the l_2 values used for the RSP-BIC (e) broadening, (f) annihilation and (g) sharpening processes. The final row (h)-(j) shows the Q-factor as a function of Δn for each of these processes.

In **Figure S7a and S7b** we plot the resonance position and the imaginary part of the hybrid eigenfrequencies γ_{rad} as functions of the perturbation Δn , obtained through numerical simulations in COMSOL Multiphysics and the RSE theory from Eqs. (7) and (8), for both RSP-BIC and PD resonances. As shown, the RSE theory accurately reproduces the full-scale numerical modeling for both real and imaginary parts of eigenfrequencies.

Introducing Δn shifts the resonance position for both resonances in the shorter wavelength region: RSP-BIC and PD resonance are shifted by 8 nm and 15 nm, respectively. The radiative part of the PD resonance depends linearly on the perturbation, while, γ_{rad} of the RSP-BIC exhibits, at first sight, an almost quadratic dependence on the perturbation, which is typical of regular BIC behavior.⁹ In **Figure S7c** we plot the Q-factor as a function of Δn on a logarithmic scale and reveal that the function depends on the perturbation in a more peculiar manner than inverse quadratic law $Q \propto 1/(\Delta\epsilon)^2$.

To explain this, we analyze the hybrid eigenfrequency of qBIC from Eq.(8). The electric fields of a true BIC are supposed to be real, which leads to a purely real parameter v_2 . However, we analyze here the RSP-BIC with a large but still finite Q-factor $Q_0 = \omega'_2/(2\gamma'_2)$ (where the complex eigenfrequency of RSP-BIC is $\omega_2 = \omega'_2 + i\gamma'_2$, so the fields of the eigenmode still have a small imaginary part. This results in a complex $v_2 = v'_2 + iv''_2$, $v'_2 \sim 10^{-2}$ and $v''_2 \sim 10^{-6}$. One may fairly point out that v''_2 can be neglected. However, as we will show below, this component contributes

to the peculiar behavior of the Q-factor and causes its deviation from the typical inverse square law. In addition to v_2 the eigenfrequency of the PD resonance ω_1 and the overlap integral u between the states also have the imaginary parts. For the sake of brevity, we introduce $\alpha = u^2\omega_2(\omega_1 - \omega_2)^{-1} = \alpha' + i\alpha''$. Then the Q-factor of the qBIC reads as:

$$Q_{qBIC} \approx \frac{1}{\frac{1}{Q_0} - 2v_2''\Delta\varepsilon - 2\alpha''(\Delta\varepsilon)^2}. \quad (9)$$

Now one can see, that:

- (i) In the case of $\Delta\varepsilon = 0$ the Q-factor of qBIC returns to the finite Q-factor Q_0 of the RSP-BIC;
- (ii) Due to the nonzero value of v_2'' , the linear term in $\Delta\varepsilon$ plays an important role at small perturbation values, and the Q-factor function deviates from the typical inverse square law;
- (iii) At the relatively large $\Delta\varepsilon$ the main contribution comes from a quadratic term with α'' , which includes a mixing parameter u between the PD resonance and the qBIC.
- (iv) In the case of a perfect BIC with $v_2'' = 0$ and $Q_0 \rightarrow \infty$ the Q-factor indeed exhibits the typical dependence on perturbation $Q \propto 1/(\Delta\varepsilon)^2$

Finally, we plot the Q-factor of RSP-BIC with $l_2^{RSP} = 210.5$ nm as a function of Δn according to the analytical expression (9) and demonstrate close match between the RSE theory and full-scale simulations.

Next, using the Eq. (8), we plot in **Figure S7d** γ_{rad} of qBIC as a function of l_2 for stationary ($\Delta n = 0$) and perturbed ($\Delta n = 0.13$) problems. According to the main text, we investigate three other different scenarios (in addition to the case of resonance creation, shown in **Figure S7a-S7c**): (i) broadening the resonance width (increase in γ_{rad} with initially nonzero radiative losses), (ii) resonance annihilation (decrease of γ_{rad} to zero), and (iii) resonance sharpening (decrease of γ_{rad} to a nonzero value). For all three cases we fix l_2 at (i) 219 nm, (ii) 204 nm and (iii) 187 nm, respectively (see arrow marks in **Figure S7d**). As before, we use the RSE theory and compare it with COMSOL results. In all three cases the RSE theory closely matches the full-scale simulations. It also predicts that the perturbation $\Delta n = 0.13$ increases γ_{rad} up to 0.12 THz for the ‘‘broadening’’ case and decreases γ_{rad} down to 0 THz or 0.20 THz for the ‘‘annihilation’’ and ‘‘sharpening’’ cases, respectively, which closely matches the experimental data fitting, showing in **Figure 5** of the main text.

One may notice that the RSE theory predicts negative γ_{rad} values for the ‘‘annihilation’’ case when $\Delta n \geq 0.14$, which has no physical meaning and demonstrates a slight overreach beyond the theory’s limits of applicability. Additionally, in **Figure S7h-j** we plot the Q-factor dependencies on the perturbation Δn in logarithmic scale.

Figure S8: Analysis of hybrid eigenstates of the metasurface with environmental perturbation $\Delta\varepsilon = (\mathbf{n}_0 + i\Delta\mathbf{k})^2 - \mathbf{n}_0^2$. Resonance position and the imaginary part of the eigenfrequencies ($\gamma_0 = \gamma_{\text{rad}} + \gamma_{\text{int}} \approx \gamma_{\text{int}}$) of the (a) RSP-BIC and (b) PD modes, obtained via COMSOL Multiphysics (solid lines) and RSE theory (dashed lines), as functions of the extinction coefficient variation Δk .

We separately consider the case where the change in $\Delta\varepsilon$ is due to the dissipation losses only. We introduce an environmental perturbation as: $\Delta\varepsilon = (\mathbf{n}_0 + i\Delta\mathbf{k})^2 - \mathbf{n}_0^2$, and obtain the expected result: Δk doesn't shift the resonance position of the RSP-BIC ($l_2^{\text{RSP}} = 210.5$ nm) and, therefore, does not introduce a radiative part γ_{rad} . At the same time, the imaginary part of the eigenfrequency $\gamma_0 = \gamma_{\text{rad}} + \gamma_{\text{int}} \approx \gamma_{\text{int}}$ depends linearly on Δk (See **Figure S8**).

As a conclusion of this part, we have clearly demonstrated:

- (i) The RSE theory can qualitatively reproduce the change in eigenfrequency due to environmental perturbation in one of the nanorods. It also shows that the introduction of perturbation mixes the antiparallel dipole qBIC and PD resonance.
- (ii) The RSP-BIC Q-factor depends on the perturbation in accordance with the Eq.(9), which is deviates from the typical inverse quadratic law.
- (iii) The agreement between the experimental data for γ_{rad} and the values predicted by the RSE theory for broadening, annihilation and sharpening cases allows us to evaluate the change in the nanorods refractive index due to pumping as $\Delta n \sim 0.1$.
- (iv) The introduction of Δk as an environmental perturbation results in a linear dispersion of the imaginary part of the modes' eigenfrequencies.

Further, we added a new Figure 4 to the main manuscript as well as describing text and a link to Figure 5 (old Figure 4):

Pumping Induced Refractive Index Perturbation and Ultrafast Tuning

To quantify the pump-induced refractive index change, we match the experimentally observed resonance shift of 9.2 nm to numerical simulations (**Supplementary Note 11**), yielding a refractive-index difference between the two rods of $\Delta n = n_{\text{rod2}} - n_{\text{rod1}} = 0.13$ (**Figure 4a**). To interpret this value analytically, we employ resonant-state expansion (RSE) theory⁸ (**Supplementary Note 5**). For simplicity, we assume a constant crystalline-silicon index $n_0 =$

3.7 and place the metasurface in vacuum. With these parameters, the RSP-BIC ($Q > 10^7$ in simulations) occurs at $l_2^{\text{RSP}} = 210.5$ nm. Treating the pump induced permittivity change as a perturbation $\Delta\varepsilon = (n_0 - \Delta n)^2 - n_0^2 \approx -2n_0\Delta n$, the RSP-BIC with complex eigenfrequency ω_2 can be hybridized following RSE theory with a parallel electric-dipole mode (ω_1). Based on the hybrid eigenfrequency ω_{qBIC} an expression for γ_{rad} can be found:

$$\gamma_{\text{rad}} = \text{Im}(\omega_{\text{qBIC}}) \approx \text{Im}\left(\omega_2 \left[1 - v_2\Delta\varepsilon - \frac{u^2\omega_2(\Delta\varepsilon)^2}{\omega_1 - \omega_2}\right]\right). \quad (2)$$

The two perturbation matrix elements v_2 and u arise from self- and cross-coupling of the RSP-BIC to the parallel-dipole mode, respectively (see **Supplementary Note 5**).

Figure 4b tests the RSE by comparing γ_{rad} from simulations (dots) with Eq. (2) (solid lines) as a function of l_2 for $\Delta n = 0$ (gray) and $\Delta n = 0.13$ (blue), resembling experimental results with and without pump. The perturbation shifts l_2^{RSP} to smaller values, creating two spectral regions: in the green shaded region γ_{rad} increases when the pump is applied, whereas in the red shaded region γ_{rad} decreases. Four representative cases are highlighted by arrow marks and corresponding Δn sweeps are shown in **Figure 4c**: Case 1 ($l_2 > l_2^{\text{RSP}}$) leads to an increase of γ_{rad} , *broadening* the mode. Case 2 ($l_2 = l_2^{\text{RSP}}$) *creates* a bright mode ($\gamma_{\text{rad}} > 0$) from a dark mode ($\gamma_{\text{rad}} \approx 0$). As the opposite to creation, case 3 (l_2 slightly smaller than l_2^{RSP}) yields a decrease of γ_{rad} toward zero, *annihilating* the resonance at $\Delta n = 0.13$. Case 4 ($l_2 < l_2^{\text{RSP}}$) *sharpens* the mode by lowering γ_{rad} while keeping it larger than zero. The close agreement of simulations and RSE theory confirms the applicability of the derived analytical model for the considered refractive index perturbation range.

Figure 4 | RSE analysis of refractive index change near the RSP-BIC. (a) Unit cell in which the refractive index of rod 1 (n_{rod1}) is reduced by Δn with respect to the constant index of rod 2 (n_{rod2}). (b) Simulations (dots) are compared with the resonant-state expansion (RSE) theory where γ_{rad} is plotted versus l_2 for $\Delta n = 0$ (gray) and for

the experimentally estimated $\Delta n = 0.13$ (blue). Increasing Δn shifts the RSP-BIC to smaller l_2 , so depending on l_2 , moving from the gray to the blue curve either increases (green shaded region) or decreases γ_{rad} (red shaded region). Four representative metasurfaces (1–4) and their corresponding γ_{rad} are shown in (c) as a function of Δn for: (1) mode broadening (γ_{rad} increases); (2) dark-to-bright creation (γ_{rad} increases from zero to a finite value); (3) annihilation (γ_{rad} decreases approaching zero); (4) sharpening of a bright mode (γ_{rad} decreases).

To experimentally explore these four switching cases, we perform pump-probe measurements at four corresponding positions along the l_2 -gradient metasurface shown in **Figure 2f-i**. **Figure 5a** sketches the expected pump-induced spectral change: the pump creates a $\Delta n = 0.13$, shifting the resonance towards shorter wavelengths. Furthermore, the RSP-BIC condition shifts to smaller l_2 in accordance to **Figure 4b**.

Reviewer comment:

Some minor technical points are listed below:

(a) The authors should use the absolute values of transmittance in the colorbar of Fig. 3e,f.

Answer & action taken:

We added the absolute values of the transmittance to the new Figure 3e.

Reviewer comment:

(b) The Supplementary Materials would benefit from short descriptions of the figures that go beyond the captions to make it more autonomous.

Answer & action taken:

We are following the Reviewer's request and embed the supplementary figures into Supplementary notes to increase the standalone readability of the SI. Note, this change is extensive throughout the Supplementary Information. Further, now only notes are referenced from the main manuscript to the Supplementary Information.

Reviewer comment:

(c) Can the authors comment on the inconsistency of the transmittance amplitude measured in linear spectra (Fig. 1h) and time-resolved spectra? This is likely connected with the major point (i) above.

Answer:

Indeed, there is a slight variation between the linear obtained spectra (shown in Figure 2) and the time-resolved spectra (shown in Figure 3-5). As noted in our methods section, the linear spectra are obtained using a collimated white light source, while the time-resolved measurements were performed using a pulsed laser. The spatial and temporal coherence of the pulsed laser can slightly alter the overall extinction and scattering through interference effects.

Reviewer comment:

(d) The discrepancy of spectral profile of the modes in Fig. 2d and h is also too obvious to completely neglect in the discussion.

Answer:

We acknowledge that there is indeed a spectral mismatch (about 10 nm) between the simulated and experimental SP-BIC modes in Figures 2d and 2h. While a 10 nm offset might appear large, we consider it to be relatively minor compared to potential uncertainties in the measured geometry. This specifically includes two factors: Size measurement uncertainty and idealized resonator shape.

Our simulation parameters were derived purely from SEM images, which inherently carry measurement uncertainties. Small inaccuracies (e.g., SEM calibration, user-dependent measurement techniques) can produce a 10 nm offset. For instance, in Figure R1 (below), we show that uniformly scaling the unit cell size down by just 1.5% can shift the mode by 15 nm, greater than the discrepancy here. E.g. this scaling change would only lead to a change of w_2 by 1.5 nm and a change of the height by 1.7 nm.

The fabricated resonators have slightly rounded corners and slightly angled sidewalls, whereas our simulations assume perfect and identical rectangular cross-sections. Even minor deviations in height or edge curvature can shift the resonance by a few nanometers.

Despite this offset, the overall agreement between simulation and experiment is strong: the key trends, the emergence of the lossy SP-BIC with increasing w_1 , and its disappearance/reappearance around the RSP-BIC condition are visible. As Figure 2h shows, the ability to map these resonances continuously with our gradient metasurface design is a significant advance compared to previous SP-BIC works, confirming that the physical phenomena of RSP-BICs are reproduced in both data and simulations.

Figure R1: Influence of Unit-Cell Scaling A unit cell equivalent to that in Figure 2d ($l_2 = 236$ nm) is uniformly scaled by a factor S . The unscaled cell (gray) has $S=1$, while the scaled cell (blue) is 1.5% smaller ($S=0.985$). Even this small change in all geometric parameters (pitch, resonator lengths, widths, and heights) causes a 15 nm spectral shift, exceeding the ~ 10 nm discrepancy between experimental and simulated results in Figure 2.

Action taken:

We added a sentence on page 5 discussing the discrepancy between simulations and experiments.

The small spectral shift between simulated and experimental results of around 10 nm can be attributed to slight geometrical offsets between simulated and measured geometries.

Reviewer comment:

(e) Fig. 3e,f seem to indicate different incident angles for pump and probe. Does this correspond to any realistic geometry? Taking into account the point (i) above, maybe this schematic could be improved.

Answer & action taken:

We thank the Reviewer for pointing out this unfavorable presentation of the pump-probe experiment.

We replace the k-vector representation in these plots which now moved to Supplementary Note 7 with two pulses (pump and probe) of normal incidence, which are the conditions that we have in the experiments.

Reviewer comment:

(f) The part of discussion related to Sb₂S₃ seems too unrelated to the core results of the paper. The utilization of phase-change materials, especially antimony compounds, has its own challenges which deserve a separate research effort. In the current state, I feel that it breaks the otherwise very smooth flow of the paper.

Answer & action taken:

We thank the Reviewer for this concern. We removed the corresponding passage from the main text and the SI.

Reviewer comment:

(g) The “femtosecond timescale” of creation and annihilation mentioned by the authors is a bit misleading, I would suggest using “sub-picosecond” instead - this does not diminish the quality and significance of the results anyways.

Answer & action taken: We agree with the Reviewer on this point. The phrase in the text has now been changed to “sub-picosecond”.

Referee #2

This manuscript presents a first-of-its-kind experimental demonstration of a resonant optical structure with an ultrafast tunable radiative decay rate of the resonant mode. The manuscript presents a highly original and elegant idea supported with a thorough experimental verification. Overall, this is a very well-executed, well written and structured manuscript which I enjoyed reading as a referee. I particularly want to emphasize the excellent presentation; the manuscript is easy to follow. That being said, I do have a number of serious concerns about the interpretation and the analysis of the experimental data, which need to be thoroughly addressed before the manuscript can go any further down the publication chain.

Answer:

We sincerely thank the Reviewer for their thoughtful and positive evaluation of our work, and we are pleased to hear that the manuscript was found to be original, well-executed, and clearly presented. We also appreciate the Reviewer's many critical comments, which we address in detail below point by point.

Reviewer comment:

1. I believe the sketch in Fig. 1(d) could be more informative if the authors show not only the annihilation of the resonance, but also illustrate its progressive narrowing/broadening, which is the key result of γ_{rad} variation.

Answer:

We agree with the Reviewer that Figure 1a-d would benefit from showing intermediate states. Especially for (d) the sharpening or broadening effect should be shown to the reader.

Action taken:

We added intermediate states to Figure 1a-d.

Reviewer comment:

2. In caption of Fig. 1(f) did the authors imply “resulting in asymmetric dipole moments and the emergence of the _quasi_ SP-BIC mode”?

Answer & action taken:

We thank the Reviewer for pointing out this misleading caption. We changed it as follows:

(f) The metasurface initially exhibits an RSP-BIC, where the dipole moments of both rods with refractive indices $n_{rod1} = n_{rod2}$ are of equal strength, resulting in an almost perfect antisymmetric mode profile with γ_{rad} approaching zero.

Reviewer comment:

3. Does the condition $p_{tot} = 0$ in an asymmetric metasurface guarantee an exact BIC with infinite lifetime, or this is just a good approximation? The SP-BIC is an exact solution of Maxwell’s equations with exactly zero radiative loss. Does that also work with the RSP-BIC? Although the total dipole moment of the unit cell is zero, I can imagine its total higher-order moments (clearly being non-zero due to the geometric asymmetry) producing a $k_x = 0$ Bloch mode with non-zero radiation.

Answer:

We thank the Reviewer for raising this point. In a conventional SP-BIC, the radiative loss γ_{rad} is equal to zero when perfect geometric symmetry is maintained (for instance, when the two rods that form the anti-parallel dipole are truly identical). For RSP-BICs, reaching $\gamma_{rad} = 0$ is more tricky and needs very fine adjustment of resonator parameters to reach this point. Nevertheless, the numerical and experimental results shown in Figure 2 utilizing a continuously tuned gradient metasurface with very smooth parameter changes clearly show the full resonance modulation functionality of our approach. Further, in the first figure of the newly added Supplementary Note 5, we show that even through a fairly coarse rastering of the w_1 and l_2 parameter space Q-

factors exceeding 10^7 can be found for a range of w_1 and l_2 configurations, which, effectively mean that the mode is almost dark with γ_{rad} approaching to zero.

Reviewer comment:

4. What exactly is the “loss density distribution” that the authors present in Fig. 3? This is the first time I’m encountering such a terminology in nanophotonics.

“integrated X loss”, “electromagnetic? loss per volume”, or maybe even better “absorbed power density distribution”

Answer & action taken:

We thank the Reviewer for pointing out this important clarification. The quantity plotted in Figure 3 is the power loss density (in units of W/m^3), which represents the energy dissipated per unit volume. We have now clarified this in the main text (page 7) and updated the caption of Figure 3 to explicitly state both the definition and the corresponding units.

...

As the active tunability is based on lowering n via photoexcitation, we compare the power loss density (energy loss per unit volume in W/m^3) distribution within both rods.

...

(b) The unit cells' normalized power loss density map (energy loss per unit volume in W/m^3) for the three Mie modes shows distinct mode profiles and dissimilar losses in both rods at a cutting plane of $z = 30 \text{ nm}$. (c) Comparison of average power loss density between the two rods for each Mie mode. Mie 1 exhibits the highest ratio between Rod 1 and Rod 2, indicating the strongest selective absorption.

...

Reviewer comment:

5. One of my key criticism relates to the results presented in Fig. 3(f) and their interpretation in the text. The authors’ interpretation of these measured spectra is “... the increase in resonance amplitude suggests a change in γ_{rad} ”. A change of γ_{rad} (in a nearly lossless system) would result in a narrowing/broadening of the resonant peak/dip/Fano resonance (depending on the background scattering matrix of the system), but its “amplitude” would remain unchanged. Such a behavior easily follows from the single-mode CMT model that the authors employ throughout their study. In Fig. 3f I do not see any such narrowing or broadening of the resonant line, but instead I do see a substantial change in the amplitude of the transmission dip, which cannot happen solely as a result of γ_{rad} variation in a lossless single-mode system.

Answer:

As the Reviewer correctly points out, changing γ_{rad} in a nearly lossless system results in a narrowing or broadening of the resonance, i.e. a change of the overall Q-factor, while maintaining near-unity transmission amplitudes. Although this change in width is also observable for us (Q decreases in the respective figure by roughly 1/4, see **Figure S17**), its effect is moderate compared with the much more striking change in amplitude in **Figure 3f**. This comes from these measurements being performed relatively close to the symmetry-protected regime, where radiative losses are small. In this undercoupled range, intrinsic losses are non-negligible and usually larger than radiative losses (see **Figure S18**). This limits the maximum Q-factor as well as the resonance amplitude, which now strongly scales with radiative losses: the minimum in transmission in eq. 1 is given by $1 - 1/(1 + \gamma_{\text{int}}/\gamma_{\text{rad}})$, meaning that any increase in γ_{rad} will increase the amplitude.

Action taken: The corresponding sentence has been changed to clarify that we are in a regime of low coupling:

...

Furthermore, after the pump, the resonance amplitude shows a clear increase, indicating a significant change in γ_{rad} due to the experiments being performed in the undercoupled regime.

...

Reviewer comment:

6. In that regard, I am very curious what the CMT fits of the measured transmission spectra look like. The main text as well as SI shows numerous plots of γ_{rad} vs time, but not a single CMT fit of the experimental data, which would be crucial to judge whether optical pump is indeed capable of modifying its purely radiative decay rate. Such fits would be very helpful for interpreting the data shown in Figs. 2(h) and (i), for example, where the transition from measured spectra to γ_{rad} occurs without showing explicitly any intermediate fitting step. Same about Figs. 4(b)-(e).

Answer:

Following the request of the Reviewer, we added Supplementary Note 9 including three new figures focusing on the case of mode broadening with $\lambda_2 = 236$ nm. Figure S14 and Figure S15 show 100 spectra obtained between -10 ps and 35 ps (shown in Figure 3e and Figure 5b) and their corresponding fits. Figure S16 presents the obtained γ_{rad} and γ_{int} data (as shown in Figure 3f) including error bars. The estimated mean error for γ_{rad} is around 2.4%, while the error for γ_{int} is 3.6%, demonstrating high fitting accuracy.

Action taken:

On page 8 we added a linking text:

Using the established TCMT model (**Supplementary Note 1**), we fit the transient spectra (**Supplementary Note 9**) and extract the time-dependent loss rates shown in **Figure 3f**.

Further, we added Supplementary Note 9

Supplementary Note 9: TCMT fitting of temporal evolution.

Extracting radiative and non-radiative losses from the measured transient absorption traces requires fitting the spectra at each time delay with the TCMT theory discussed in Supplementary Note 1. **Figures S14** and **S15** show the good convergence of the fit to the experimental data, which results in low relative errors (**Figure S16**).

Figure S14: Temporal evolution of spectra and TCMT fits 1. Experimental spectra (black), and TCMT fits (red) for the time evolution from -10 ps to 6.7 ps, for $l_2 = 236$ nm. The data set is part of the time trace shown in **Figure 3e** and **Figure 5b**.

Figure S15: Temporal evolution of spectra and TCMT fits 1. Experimental spectra (black), and TCMT fits (red) for the time evolution from 7 ps to 35 ps, for $l_2 = 236$ nm. The data set is part of the time trace shown in **Figure 3e** and **Figure 5b**.

Figure S16: Standard errors of fitted data. Standard errors obtained for (a) γ_{rad} , and (b) γ_{int} using the TCMT model and the non-linear least-squares minimization package lmfit. The mean standard error for γ_{rad} is 0.0019 (2.4%), while the mean error for γ_{int} is 0.039 (3.6%).

Reviewer comment:

7. There is an argument to be made that, instead of considering the unit cell as a single entity with some “black box” CMT parameters, one could consider some exact metasurface unit cell as a combination of two separate particles supporting their own Mie-like resonant states, which couple in the near field by the dipole-dipole interaction. From that perspective, the effect of optical pump manifests as lowering Si refractive index, which in turn leads to the familiar spectral resonance shift of one of the two nanoparticles in the dimer. Which is exactly what the authors want to distance themselves by considering the dimer as a single black box with some resonant characteristics. That is to say, whether this work demonstrates tuning of the resonant frequency or the radiative decay rate depends on the perspective we analyze the system from.

Answer:

We thank the Reviewer for this remark – as they correctly point out, we are in fact changing optical properties of one of the two rods, which (in the absence of the other rod) would just result in a spectral shift of its Mie resonance. However, when investigating the quasi-BIC mode, the presence of both particles is crucial, as it does not exist in a system only containing one particle, but instead is localized in both of them (see Figure S2 and S6d). In this system, far-field transmittance spectra cannot reveal these two modes of isolated rods since the resonators are sub-wavelength spaced. We therefore avoid the description of the coupled system purely by its constituents.

Reviewer comment:

8. Yet another piece of information that this work crucially lacks is any theoretical description of the refractive index modulation by an optical pump. While the authors do not focus on the refractive index-based description of the resonance modification and prefer to describe its evolution in terms of CMT decay rates, providing even a simplest model of refractive index modulation would be helpful in two ways: (i) it would highlight the key mechanism enabling index modulation, and (ii) it could perhaps be valuable in analyzing the transient dynamics of the resonant characteristics shown in Fig. 5. Perhaps something like two-photon absorption + a bi-exponential charge carriers density decay would be sufficient for such a simple model.

Answer and action:

We appreciate the Reviewer's suggestion and have acted on it. A quantitative estimate of the pump-induced refractive-index change is now presented in Supplementary Note 11 (see also our reply to Reviewer 1, point iv). In addition, we introduce a resonant-state-expansion (RSE) framework that links this Δn to the observed shifts in eigenfrequency and γ_{rad} ; the model is summarised in the new Figure 4 and detailed in Supplementary Note 5. We hope the Reviewer will find this expanded analysis helpful and interesting.

We further add Supplementary Note 10 (reprinted in our answer to Reviewer 1, comment ii) discussing in detail how the resonance evolves over time. As the Reviewer correctly points out, a bi-exponential model would be a good description of decay dynamics. However, due to the small observed impact of Auger recombination, we find that the refractive index is almost exclusively described by a single exponential decay (see **Figure R2**). To avoid overfitting, we restrict the analysis to this single decay related to (surface) carrier recombination, with a time constant of 20 ps.

Figure R2: Evolution of the resonant spectral position at the resonance broadening condition ($l_2 = 236$ nm) and a pump fluence of $100 \mu\text{J}/\text{cm}^2$. The fit shows a good agreement of the decay with a single exponential, at a time constant of 16 ps. At low pump fluences, the decay time converges to 20 ps.

Reviewer comment:

9. If I'm not mistaken, Fig. 5 is the first instance where γ_{int} comes into the analysis of the optical system response for the first time. Quite surprisingly, the authors' analysis shows that γ_{int} does increase substantially during pumping. This could actually address the important point I was making in Comment 5. But if it does, the analysis performed for the data in Figs. 3 and 4 should then incorporate this γ_{int} (which at the moment it does not, I believe).

Answer:

We thank the Reviewer for raising this concern. Reviewer 1 has a similar question in point (iv), and we refer the Reviewer to our detailed discussion in that section, where we estimate the change in the real and imaginary parts of the refractive index using a combination of simulations and experimental data. In addition, we have added several clarifications (see our response to Reviewer 1) indicating the change in γ_{int} switching, including a reference to this as early as in the introduction.

Reviewer comment:

10. Page numbers in many of the references are either missing, or incorrect (the ones starting with page "1" for sure). Many of the references even miss journal names.

Answer: We updated the references.

Reviewer comment:

To conclude, this is an interesting, highly original, innovative and significant work for the field of nanophotonics and electromagnetism in general, which could be considered for publication in Nature if the authors do accurately address all the issues related to the analysis and interpretation of the experimental data.

Answer:

We again thank the Reviewer for their positive evaluation of our work, and we hope our changes fulfill the Reviewer's expectations.

Referee #3

Reviewer comment:

I co-reviewed this manuscript with one of the Reviewers who provided the listed reports.

Answer:

We thank the reviewer for their co-reviewing.

Referee #4

Reviewer comment:

In this work, Aigner, Possmayer and co-authors demonstrate, for the first time to my knowledge, the ultra-fast tuning of the radiative loss of a metasurface system. Compared to previous works, the main novelty lies precisely in that they deal primarily with the radiative part of the losses, by which they can create, annihilate and modulate the width of a resonance, namely a (quasi-)bound state in the continuum (BIC). Indeed, previous works that demonstrated tuning of similar type of modes in a metasurface dealt mainly with modulations of either its spectral position, its non-radiative (absorption) losses, or both. This radiative-loss tuning is here achieved using a clever idea. To illustrate it, first, the authors break the geometrical symmetry of the unit cell without breaking the optical symmetry, forming what they refer to a restored symmetry protected BIC (RSP-BIC). This mode exhibits a full cancellation of the net dipole moment despite being formed by two dissimilar nanorods. Interestingly, this opens the possibility of selectively pumping one of these rods, therefore being able to modulate its dipole moment. This is used to turn this RSP-BIC into a qBIC that can couple to external radiation (creation mode). The same concept is then used to turn into a (true) RSP-BIC a mode that was initially a qBIC one (annihilation mode). Partial states that induce sharpening and broadening of the resonance can also be accessed with slightly modified conditions.

While I have no doubts about the novelty of the approach, which is also creative and (as the authors point out) can be extended to other scenarios, I have more doubts about its significance. In particular, the authors claim that their results open “new possibilities for studying light-matter coupling effects[...]. This is in contrast to conventional ω_0 and γ_{int} tuning methods, which struggle to dynamically control local field enhancement or spectral width without introducing significant parasitic losses”. In my opinion, however, the authors fail to show evidence for any of such scenarios. Can they show/illustrate one case in which this method yields effects not attainable by either ω_0 and/or γ_{int} tuning?

Answer:

We thank the Reviewer for acknowledging the novelty and creative approach of our work. To address the concern regarding the broader significance of γ_{rad} tuning and its advantages over tuning ω_0 or γ_{int} , we have added a new Supplementary Note 12 discussed in our response to Reviewer 1 that outlines six specific examples where γ_{rad} control offers clear benefits. We would particularly like to draw attention to points 3 and 4 (polaritonic critical coupling and tunable quantum light emission), which directly relate to the Reviewer’s remark on “new possibilities for studying light-matter coupling effects.”

Action taken:

We added the following sentences in the introduction on pages 2 and 3 to show, besides the novelty of γ_{rad} tuning, its potential broad impact on active photonics:

This capability is transformative for active photonics, particularly in applications like optical communications, signal processing, or filtering, where it can minimize spectral crosstalk and avoids parasitic losses. Furthermore, in contrast to ω_0 and γ_{int} , γ_{rad} directly governs the resonance amplitude, Q-factor, and local field enhancement, making it critical for precise control of metasurface functionalities.

...

Thus, the ability to arbitrarily tune radiative loss to increase or decrease the Q-factor and achieve on/off switching of resonances has so far not been achieved, impeding future applications of metasurfaces like telecommunication switches and optical filters.

We added the following section in the discussion on page 11:

In addition to enabling low-loss, all-optical switching for telecommunications or computing, this direct control over radiative coupling offers significant advantages for time-resolved enhanced light-matter interactions such as quantum emission and polariton-based effects.

Further, we added a more detailed discussion on various possible applications demonstrating the broad impact of our approach in a new supplementary note linked on page 11:

The ability of γ_{rad} to precisely control resonance cavity parameters (e.g., field enhancement and Q-factor) as well as being able to toggle between a resonant and nonresonant system offers many possibilities throughout active nanophotonics, with several key examples detailed in **Supplementary Note 12**.

Further, we added Supplementary Note 12

Supplementary Note 12: Potential applications for radiative loss based active photonics

The possible applications for γ_{rad} tuning are diverse. Below, we specifically name six possible directions where γ_{rad} tuning can provide improved performance and new functionalities compared with conventional approaches based on tuning ω_0 or γ_{int} .

1. On/Off Switchable Filters: Conventional approaches to resonance tuning, i.e., shifting ω_0 or increasing γ_{int} , cannot fully switch a resonance on or off. ω_0 tuning moves the mode away from its original spectral position to reduce undesired reflection at that specific wavelength, but it cannot create a completely transparent metasurface system because a highly reflective resonant mode is always present. γ_{int} tuning increases intrinsic losses and thus quenches the reflectance amplitude of resonance but still leaves some broadened residual mode, as the structure remains radiatively coupled, just with smaller amplitude and broader linewidth. Further, it typically leads to overall lower transmission. In contrast, γ_{rad} tuning can entirely decouple the mode from the far field using a system that transitions from truly “resonance-free” ($\gamma_{\text{rad}} = 0$) to fully “resonant” ($\gamma_{\text{rad}} > 0$). The result is an on/off switchable filter that, in its “off” state, is truly transparent, minimizing spectral crosstalk and absorption. This capability is important for advanced active optical filtering in photonic circuits.

2. On-Demand Sensors: In sensing applications, the ability to activate or deactivate a resonance at will can help to keep an optical system transparent and only switch on resonant modes temporally for sensing if needed. ω_0 tuning leads to resonant modes that are always present, and thus, for some wavelengths, the system is highly reflective. γ_{int} tuning introduces losses into the system and reduces optical transparency for resonant and off-resonant wavelengths. γ_{rad} -driven sensors can remain fully transparent in “off” mode, transmitting the entire spectral band, and then selectively switch “on” the resonance only when particular analytes need to be detected. This is especially advantageous in fiber-based optical systems, where the sensor element remains transparent during normal operation and can be optically activated to create a high-Q response for refractive index sensing.

3. Polaritonic Critical Coupling: Light-matter coupling strength for polariton formation depends on the balance between photonic and material losses. Typically, the photonic mode’s total loss $\gamma_{\text{photon}} = \gamma_{\text{rad}} + \gamma_{\text{int}}$ should match or be similar to the material excitation’s linewidth γ_{mat} to reach so-called polaritonic critical coupling.² Thus, there is a need to actively tune the photonic loss rates to probe and optimize this ratio. ω_0 tuning merely shifts the mode’s center frequency and cannot reduce or increase the total loss channel. γ_{int} tuning, meaning increasing intrinsic cavity losses, quenches near-field enhancement and can outweigh the material excitation, effectively preventing polariton formation. The γ_{rad} tuning advantage lies in keeping γ_{int} low and only altering the radiative part of the photonic loss so that it matches the material’s loss.

4. Emission Linewidth Control and Tunable Lasing: Many photonic devices rely on controlling the emission properties, such as linewidth and coherence, of an integrated gain medium. In many applications, there is a need to actively control these properties. ω_0 tuning shifts the resonant frequency of the photonic mode but does not fundamentally alter its damping, so the emission linewidth cannot be changed. γ_{int} tuning induces additional intrinsic losses and reduces the overall emission strength, as further nonradiative decay channels are introduced. γ_{rad} tuning preserves the low nonradiative intrinsic losses of the photonic mode while allowing the radiative decay rate and thus the linewidth to be freely adjusted. For lasing applications, the threshold condition depends on the total cavity losses, and if γ_{rad} is too low, it can be difficult to couple pump light into the cavity or out-couple coherent photons, while if γ_{rad} is too high, the threshold power increases. Dynamically controlling γ_{rad} allows optimization of the cavity's Q-factor, enabling broad- and narrow-line lasing modes with adjustable thresholds.

5. Ultrafast Pulse Modulation and Time-Crystal Metasurfaces:

In the spectral vicinity of the quasi-BIC, significant dispersion is induced on propagating light. γ_{rad} based resonance tuning therefore leads to far-reaching control over key material properties on ultrafast timescales, which is a requisite for experimentally achieving time-crystal metasurfaces. Furthermore, it can be used for novel compact pulse-compression systems: By modifying the dispersion on the same timescale as the pulse, a temporally varying dispersion is induced, meaning the front of the pulse propagates at a different speed than its back. Therefore, this scheme controls pulse lengths without splitting them into their spectral components, potentially allowing on-chip solutions.

Compared with γ_{int} tuning, this approach offers two main advantages: First, tuning γ_{rad} enables precise control over how long light stays in the cavity or when it leaks; γ_{int} -based tuning will only destroy the mode by increasing losses, which is unwanted in these applications. Furthermore, the measured metasurface allows for sub-picosecond increases and decreases in γ_{rad} depending on the exact geometry, hence leading to positive or negative tunable delays. For γ_{int} , on the other hand, ultrafast decreases remain elusive.

6. Nonvolatile Encoding of Metasurfaces: Phase change material-based encoding in metasurfaces can be used to store data or to adjust resonance parameters post-processing. Challenges with ω_0 or γ_{int} tuning arise because using phase-change materials like GST or Sb_2Se_3 for conventional tuning either shifts the resonance or introduces additional absorption, never fully toggling the mode from nonresonant (transparent) to resonant. γ_{rad} tuning, on the other hand, allows straightforward encoding. Starting with a metasurface in its dark state and using a localized optical pulse to partially change the phase in certain unit cells, resonances can be coupled to the farfield in a controlled manner. A subsequent erase step, for example, uniform heating, resets the material to its transparent state. Because γ_{rad} tuning can achieve a genuine “off” resonance, this leads to reconfigurable devices that stay transparent until selectively written, enabling optical storage, adjustable patterning of wavefronts, or post-processing optimization.

Reviewer comment:

As shown in Fig. 4, all the effects shown (creation, annihilation, broadening and sharpening) have also some associated (temporal) spectral shifts. Intuitively, I would think that this is because these effects are still a consequence of a spectral shift, the only difference being that, in this case, the spectral shift happens to the modes of only one of the rods forming the unit cell. Is this the case? If so, what is the Δn that one needs to consider to explain the effects?

Answer:

Indeed, alongside the change in γ_{rad} , we observe a resonance shift of up to 10 nm. This shift is caused by a reduction in the real part of the refractive index now discussed in the newly added Supplementary Note 11. However, the observed effects of resonance creation, annihilation, broadening, and sharpening are not caused by this spectral shift in ω_0 , but can instead be clearly attributed to the change in γ_{rad} . We can confidently exclude any effects coming from changes to the unit cell dimensions or its composition of two rods as asked by the Reviewer. We invite the Reviewer to refer to our answer to Reviewer 2, question 7 where we discuss a similar issue.

Further, we introduce a new model based on resonant state expansion theory in Supplementary Note 5 (also discussed in Reviewer 1 response iv). There, we clearly found a changing γ_{rad} when Δn changes.

Action taken:

We added a Supplementary Note 11 including two figures to discuss the change of n and k while pumping.

Supplementary Note 11: Estimated refractive index changes

To estimate how both the real (n) and imaginary (k) parts of the refractive index change under laser pumping, we reproduced the mode shift and broadening observed in **Figures 4b** and **5a,b** through numerical simulations. Based on **Figure 3c**, the absorbance, and thus the changes in n and k , is approximately 3.5 times higher in Rod 1 than in Rod 2 once pumped, as illustrated in **Figure S22a**.

Step 1: Estimating Δn : We first determine the change in the real part of the refractive index from the shift of the resonance wavelength. In the simulations, silicon is assigned a baseline $n = 3.67$ and $k = 0.006$ (following Schinke et al.⁷ around 790-800 nm). To precisely match the unpumped experimental resonance wavelength (i.e., $t = -1$ ps) with the simulated resonance wavelength, we slightly reduce the in-plane dimensions of the unit cell by 2.3%. **Figure S22b** shows the comparison between the experimental (dashed gray curve) and simulated (solid gray curve) spectra. Next, we tune Δn_{rod1} (and hence $\Delta n_{\text{rod1}} \approx 3.5 \Delta n_{\text{rod2}}$) until the pumped resonance

wavelength at $t = 1$ ps spectrally matches with the experimental results (blue curves). This causes a total shift of about 10 nm, which we reproduce numerically using $\Delta n_{\text{rod1}} = -0.18$. Sweeping Δn_{rod1} from 0 to -0.18 continuously shifts the resonance and simultaneously increases its amplitude, consistent with rising radiative loss γ_{rad} as shown in **Figure S22c**.

Step 2: Estimating Δk : With $\Delta n_{\text{rod1}} = -0.18$ ($\Delta n_{\text{rod2}} = -0.05$, hence $\Delta n = |\Delta n_{\text{rod1}} - \Delta n_{\text{rod2}}| = 0.13$) fixed, we estimate the accompanying change in the imaginary part of the refractive index. Because the simulation only accounts for intrinsic material loss, we first determine an effective $k = k_{\text{est}}$ that reproduces the pre-pump non-radiative loss γ_{int} measured experimentally (0.94 at $t = -1$ ps, **Figure S18**). Sweeping the same k in both rods (**Figure S23a,b,c**) shows γ_{int} varying linearly with k while γ_{rad} is unaffected; a linear fit yields $k_{\text{est}} = 0.022$ (black line), up from the literature Si value $k = 0.006$ (Schinke et al. ⁷). To find the pump-induced change we assume linear absorption and the same 3.5:1 imbalance as for Δn , i.e. $\Delta k_{\text{rod1}} = 3.5\Delta k_{\text{rod2}}$ and $\Delta k = \Delta k_{\text{rod1}} - \Delta k_{\text{rod2}}$ (**Figure S23a**). Using the post-pump value $\gamma_{\text{int}} = 1.04$ at $t = 1$ ps (**Figure 3e**), the linear fits in **Figure S23d,e** give $\Delta k_{\text{rod1}} \approx 0.004$ (blue line, $\Delta k \approx 0.003$). Thus, the pump increases k by only a few 10^{-3} , consistent with our observation that the index contrast is dominated by Δn , while Δk contributes only a minor fraction of the total modulation.

Figure S22: Estimating Δn . (a) From **Figure 3c**, pumping leads to a 3.5 times higher absorbance in Rod 1 than in Rod 2, leading to $\Delta n_{\text{rod1}} \approx 3.5 \Delta n_{\text{rod2}}$. (b) Experimental spectra at $t = -1$ ps and $t = 1$ ps (dashed gray and blue, respectively) compared to the simulations (solid gray and blue). The in-plane dimensions of the simulated structure are reduced by 2.3% to align spectrally with the unpumped experimental mode. Next, Δn_{rod1} is swept from 0 to -0.18 to match with the pumped resonance wavelength. (c) Full Δn_{rod1} sweep from 0 to -0.18 , showing a continuous 10 nm spectral shift and an amplitude increase matching the experiment.

Figure S23: Estimating Δk from pump-probe data. (a) Schematic: besides an initial k value in both rods the pump generates additional absorption with a 3.5:1 imbalance, so $\Delta k_{\text{rod1}} \approx 3.5 \Delta k_{\text{rod2}}$. (b,c) Pre-pump calibration. Sweeping the intrinsic index k from the literature value 0.006 (grey line) to 0.10 leaves γ_{rad} unchanged (b) but increases γ_{int} linearly (c), matching the experimental $\gamma_{\text{int}} = 0.94$ at $t = -1$ ps (**Figure 3e**) gives $k_{\text{est}} = 0.022$ (black line). (d,e) Post-pump estimate. With k fixed at 0.022, Δk_{rod1} is swept from 0 to 0.025. Again γ_{rad} is unaffected (d), while γ_{int} rises linearly (e); the experimental $\gamma_{\text{int}} = 1.04$ at $t = 1$ ps (**Figure 3e**) is met for $\Delta k_{\text{rod1}} \approx 0.004$ (blue line), corresponding to $\Delta k \approx 0.003$.

ability to arbitrarily tune

We added a linking text on page 7.

To quantify the pump-induced refractive index change, we match the experimentally observed resonance shift of 9.2 nm to numerical simulations (**Supplementary Note 11**), yielding a refractive-index difference between the two rods of $\Delta n = n_{\text{rod2}} - n_{\text{rod1}} = 0.13$ (**Figure 4a**).

We added a short section discussing the limiting factor on the modulation depth in Supplementary Note 3

Future improvements to increase Q-factors and modulation strength would consist of reducing surface roughness by increasing the acceleration voltage of the e-beam, increase the array size to reduce edge leakage, replace the SiO₂ layer by an even thicker sapphire layer, or shift the resonant mode towards the near infrared to reduce material absorption.

Furthermore, we added Figure 4 and a corresponding section to the main text, as well as Supplementary Note 5 on the RSE model.

Further, we added a new Figure 4 to the main manuscript as well as describing text and a link to Figure 5 (old Figure 4):

Pumping Induced Refractive Index Perturbation and Ultrafast Tuning

To quantify the pump-induced refractive index change, we match the experimentally observed resonance shift of 9.2 nm to numerical simulations (**Supplementary Note 11**), yielding a refractive-index difference between the two rods of $\Delta n = n_{\text{rod2}} - n_{\text{rod1}} = 0.13$ (**Figure 4a**). To interpret this value analytically, we employ resonant-state expansion (RSE) theory⁸ (**Supplementary Note 5**). For simplicity, we assume a constant crystalline-silicon index $n_0 = 3.7$ and place the metasurface in vacuum. With these parameters, the RSP-BIC ($Q > 10^7$ in simulations) occurs at $l_2^{\text{RSP}} = 210.5$ nm. Treating the pump induced permittivity change as a perturbation $\Delta\varepsilon = (n_0 - \Delta n)^2 - n_0^2 \approx -2n_0\Delta n$, the RSP-BIC with complex eigenfrequency ω_2 can be hybridized following RSE theory with a parallel electric-dipole mode (ω_1). Based on the hybrid eigenfrequency ω_{qBIC} an expression for γ_{rad} can be found:

$$\gamma_{\text{rad}} = \text{Im}(\omega_{\text{qBIC}}) \approx \text{Im}\left(\omega_2 \left[1 - v_2\Delta\varepsilon - \frac{u^2\omega_2(\Delta\varepsilon)^2}{\omega_1 - \omega_2}\right]\right). \quad (2)$$

The two perturbation matrix elements v_2 and u arise from self- and cross-coupling of the RSP-BIC to the parallel-dipole mode, respectively (see **Supplementary Note 5**).

Figure 4b tests the RSE by comparing γ_{rad} from simulations (dots) with Eq. (2) (solid lines) as a function of l_2 for $\Delta n = 0$ (gray) and $\Delta n = 0.13$ (blue), resembling experimental results with and without pump. The perturbation shifts l_2^{RSP} to smaller values, creating two spectral regions: in the green shaded region γ_{rad} increases when the pump is applied, whereas in the red shaded region γ_{rad} decreases. Four representative cases are highlighted by arrow marks and corresponding Δn sweeps are shown in **Figure 4c**: Case 1 ($l_2 > l_2^{\text{RSP}}$) leads to an increase of γ_{rad} , *broadening* the mode. Case 2 ($l_2 = l_2^{\text{RSP}}$) *creates* a bright mode ($\gamma_{\text{rad}} > 0$) from a dark mode ($\gamma_{\text{rad}} \approx 0$). As the opposite to creation, case 3 (l_2 slightly smaller than l_2^{RSP}) yields a decrease of γ_{rad} toward zero, *annihilating* the resonance at $\Delta n = 0.13$. Case 4 ($l_2 < l_2^{\text{RSP}}$) *sharpens* the mode by lowering γ_{rad} while keeping it larger than zero. The close agreement of simulations and RSE theory confirms the applicability of the derived analytical model for the considered refractive index perturbation range.

Figure 4 | RSE analysis of refractive index change near the RSP-BIC. (a) Unit cell in which the refractive index of rod 1 ($n_{\text{rod}1}$) is reduced by Δn with respect to the constant index of rod 2 ($n_{\text{rod}2}$). (b) Simulations (dots) are compared with the resonant-state expansion (RSE) theory where γ_{rad} is plotted versus l_2 for $\Delta n = 0$ (gray) and for the experimentally estimated $\Delta n = 0.13$ (blue). Increasing Δn shifts the RSP-BIC to smaller l_2 , so depending on l_2 , moving from the gray to the blue curve either increases (green shaded region) or decreases γ_{rad} (red shaded region). Four representative metasurfaces (1–4) and their corresponding γ_{rad} are shown in (c) as a function of Δn for: (1) mode broadening (γ_{rad} increases); (2) dark-to-bright creation (γ_{rad} increases from zero to a finite value); (3) annihilation (γ_{rad} decreases approaching zero); (4) sharpening of a bright mode (γ_{rad} decreases).

To experimentally explore these four switching cases, we perform pump-probe measurements at four corresponding positions along the l_2 -gradient metasurface shown in **Figure 2f-i**. **Figure 5a** sketches the expected pump-induced spectral change: the pump creates a $\Delta n = 0.13$, shifting the resonance towards shorter wavelengths. Furthermore, the RSP-BIC condition shifts to smaller l_2 in accordance to **Figure 4b**.

References

1. Yang, Z. *et al.* Ultrafast Q-boosting in semiconductor metasurfaces. *Nanophotonics* **13**, 2173–2182 (2024).
2. Weber, T. *et al.* Intrinsic strong light-matter coupling with self-hybridized bound states in the continuum in van der Waals metasurfaces. *Nat. Mater.* **22**, 970–976 (2023).
3. Maier, S. *Plasmonics: fundamentals and applications*. (Springer Science & Business Media, 2007).
4. Bristow, A. D., Rotenberg, N. & van Driel, H. M. Two-photon absorption and Kerr coefficients of silicon for 850–2200nm. *Appl. Phys. Lett.* **90**, (2007).
5. Fan, K., Shadrivov, I. V., Miroshnichenko, A. E. & Padilla, W. J. Infrared all-dielectric Kerker metasurfaces. *Opt. Express* **29**, 10518 (2021).
6. Liu, Y.-H. & Cheng, C.-W. The Experimental and Modeling Study of Femtosecond Laser-Ablated Silicon Surface. *J. Manuf. Mater. Process.* **7**, 68 (2023).
7. Schinke, C. *et al.* Uncertainty analysis for the coefficient of band-to-band absorption of crystalline silicon. *AIP Adv.* **5**, (2015).
8. Gorkunov, M. V., Antonov, A. A., Mamonova, A. V., Muljarov, E. A. & Kivshar, Y. Substrate-Induced Maximum Optical Chirality of Planar Dielectric Structures. *Adv. Opt. Mater.* **13**, (2025).
9. Koshelev, K., Lepeshov, S., Liu, M., Bogdanov, A. & Kivshar, Y. Asymmetric Metasurfaces with High-Resonances Governed by Bound States in the Continuum. *Phys. Rev. Lett.* **121**, 193903 (2018).

Response to the Referees

Referee #1

Reviewer comment:

I greatly appreciate the outstanding effort the authors put in the revision. The revised version is better positioned against state-of-the-art, and the arguments the authors provide regarding the advantages of their approach over structured illumination are valid and convincing. This changes the (at least, my subjective) perspective on the novelty of the paper. The new supplementary section covering the possible applications is a very sound contribution to the impact of the paper, and other raised concerns were addressed in full. I therefore support the publication of the revised paper in Nature, with a few minor points mentioned below to be taken care of.

Answer:

We thank the reviewer for their kind words and for supporting the publication of our revised manuscript! Our responses to the remaining minor points are given below.

Reviewer comment:

I recommend still explicitly mentioning the original imbalance of radiative and internal losses of the system (their absolute values, as in Fig. S16) in the main text when discussing Fig. 3f - the percentage change is indeed insightful, but does not give the full understanding of the dynamics.

Answer and action taken:

We thank the reviewer for raising this point and now mention the absolute loss values on page 5 of the revised manuscript as follows:

“In absolute terms, γ_{int} increases from around 0.9 to 1 THz while γ_{rad} increases from 0.04 to 0.14 THz.”

Reviewer comment:

Maybe the authors could briefly mention if they expect a considerable chirp for the probe pulse as it passes through multiple dispersive elements in the experimental setup and whether it could contribute to the extracted temporal dynamics for different resonance manipulation scenarios.

Answer and action taken:

We thank the reviewer for this suggestion and now discuss chirps of the probe pulse on page 16 of the revised manuscript as follows:

“Within the narrow spectral range investigated for the fabricated structures, different probe arrival times for the spectral components due to chirp induced by the setup were negligible.”

Reviewer comment:

I advise the authors to do a careful proofreading of SI, as there are several typos and errors in equations (in particular, line 452).

Answer and action taken:

We thank the reviewer for the careful reading of our manuscript and have corrected the typos in the Supplementary Information.

Referee #2**Reviewer comment:**

I would like to thank the authors for their accurate revision and the detailed response to the referees' comments.

The authors have done extensive changes to the manuscript. Particularly, CMT fits of the measured spectra presented now in SI indeed confirm the reported behavior of radiative and non-radiative decay rates. I have no further technical comments and can recommend this manuscript for publication.

Answer:

We sincerely thank the reviewer for their help in improving our manuscript and for recommending its publication.

Referee #3**Reviewer comment:**

I co-reviewed this manuscript with one of the reviewers who provided the listed reports.

Answer:

We thank the reviewer for their efforts.

Referee #4**Reviewer comment:**

The authors have made a great effort in replying to all the questions and concerns raised by all the referees. I value the extra work that they have put on their reply, both adding new technical portions that helped clarified the interpretation of the observed effects, as well as new insights that help highlighting the significance of their work. In my opinion, the work can now be published in the journal.

Answer:

We sincerely thank the reviewer for their kind words and for recommending the acceptance of our manuscript.